# S6K1-mediated phosphorylation of PDK1 impairs AKT kinase activity and oncogenic functions

Qiwei Jiang[1,2,4], Xiaomei Zhang[2,4], Xiaoming Dai[3,4], Shiyao Han[2], Xueji Wu[2], Lei Wang[2], Wenyi Wei [3], Ning Zhang[1], Wei Xie [2✉] & Jianping Guo [2✉]

Functioning as a master kinase, 3-phosphoinositide-dependent protein kinase 1 (PDK1) plays a fundamental role in phosphorylating and activating protein kinases A, B and C (AGC) family kinases, including AKT. However, upstream regulation of PDK1 remains largely elusive. Here we report that ribosomal protein S6 kinase beta 1 (S6K1), a member of AGC kinases and downstream target of mechanistic target of rapamycin complex 1 (mTORC1), directly phosphorylates PDK1 at its pleckstrin homology (PH) domain, and impairs PDK1 interaction with and activation of AKT. Mechanistically, S6K1-mediated phosphorylation of PDK1 augments its interaction with 14-3-3 adaptor protein and homo-dimerization, subsequently dissociating PDK1 from phosphatidylinositol 3,4,5 triphosphate (PIP$_3$) and retarding its interaction with AKT. Pathologically, tumor patient-associated *PDK1* mutations, either attenuating S6K1-mediated PDK1 phosphorylation or impairing PDK1 interaction with 14-3-3, result in elevated AKT kinase activity and oncogenic functions. Taken together, our findings not only unravel a delicate feedback regulation of AKT signaling via S6K1-mediated PDK1 phosphorylation, but also highlight the potential strategy to combat mutant *PDK1*-driven cancers.

[1] Department of Gastroenterology, the First Affiliated Hospital, Sun Yat-sen University, Guangzhou, Guangdong, China. [2] Institute of Precision Medicine, the First Affiliated Hospital, Sun Yat-sen University, Guangzhou, Guangdong, China. [3] Department of Pathology, Beth Israel Deaconess Medical Center, Harvard Medical School, Boston, MA, USA. [4]These authors contributed equally: Qiwei Jiang, Xiaomei Zhang, Xiaoming Dai. ✉email: Xiew56@mail.sysu.edu.cn; Guojp6@mail.sysu.edu.cn

Protein kinases, serving as the central nodes for cell signaling transduction, temporally and spatially phosphorylate diverse downstream substrates to catalytically transmit signals derived from extra- or intra-cellular alterations[1,2]. AGC kinase subfamily has been well considered as one of the most important hubs in response to growth factors, such as insulin, IGF and PDGF[3]. Given that the full activation of most AGC kinases requires the phosphorylation modification both in the activation domain (T-loop) and hydrophobic motif[4], PDK1 has been identified as the unique kinase to catalyze the T-loop phosphorylation of at least 23 AGC kinases, including AKT, S6K, serum/glucocorticoid regulated kinase (SGK), protein kinase C (PKC) and polo-like kinase 1 (PLK1)[5]. Specifically, the phosphorylation event occurred in the hydrophobic motif is primarily identified for PDK1 interaction with and phosphorylation of most of the AGC kinases, including S6K, RSK and SGK[4]. By contrast, PDK1-mediated AKT phosphorylation and activation largely depends on the activation of phosphoinositide-3-kinase (PI3K) and the generation of PIP3, which simultaneously recruits both AKT and PDK1 onto the plasma membrane, where PDK1 phosphorylates AKT in close proximity[6–8]. As a result, PDK1-mediated signaling transduction from PI3K to AKT in response to growth factors has been extensively investigated[9], however, the upstream regulation mechanism of PDK1 is still poorly understood.

PIP3 is generated by the class I PI3K in response to ligand (such as insulin and IGF)-mediated receptor tyrosine kinase (RTK) activation, and serves as a second messenger to induce the plasma membrane translocation and activation of various downstream effectors[10]. As one of the most important PIP3-dependent activation proteins, the AKT kinase binds PIP3 through its N-terminal PH domain, and is subsequently phosphorylated by PDK1, which also binds to PIP3 with its C-terminal PH domain[7,10]. On the other hand, the mTORC2 kinase could also phosphorylate the AKT kinase by anchoring PIP3 via the PH domain derived from its unique component, SAPK-interacting protein 1 (SIN1)[11]. Thus, the generation of PIP3 is a critical step for recruiting AKT and its upstream kinases to fully activate AKT kinase. Conceivably, the aberrant accumulation of PIP3 or dysregulating the binding affinity of PIP3 with the PH domain derived from PDK1 or SIN1, will potently influence AKT kinase activity. Meanwhile, the presence of copper, also contributes to AKT activation by enhancing the interaction of PDK1 and AKT[12]. Of note, the amplification/gain-of-function mutations of *PIK3CA* or deletion/loss-of-function mutations of its negative regulator, the *phosphatase and tensin-like protein* (*PTEN*) phosphatase, could dramatically elevate the accumulation of PIP3 and markedly contribute to AKT activation and oncogenic functions in a large portion of human cancers[9]. Hence, potential modifications of PDK1 on its PH domain will also possibly affect its function for interacting with PIP3 and consequently modulate AKT kinase activity.

Notably, knocking out *Pdk1* or knocking in the PIP3 interaction-deficient mutant form of *Pdk1* resulted in decreasing mouse body size, at least in part by compromising AKT kinase activity[13,14]. Emerging studies also indicate that PDK1 could induce cell motility and migration in both PH domain and PIF-binding pocket dependent manners, partially by interplaying with CDC42 binding protein kinase alpha (MRCKα)[15], or antagonizing Rho-associated protein kinase 1 (ROCK1) functions[16,17]. Importantly, PDK1 has also been evaluated to promote cancer stem cell initiation and maintenance through the phosphorylation and regulation of PLK1/Myc proto-oncogene protein (MYC) pathway[18]. In cancers, PDK1 activation has also been associated with its protein levels due to the amplification of the 16p13.3 locus, harboring the *PDPK1* gene[19], or mutations of the tumor

suppressor *Speckle-type POZ protein* (*SPOP*) in prostate cancer setting[20]. However, PDK1 displays a constitutively active feature due to the auto-phosphorylation at Ser241 located in its T-loop region[21], resulting in the poorly investigating on its upstream regulation. Here we report that, as a bona fide substrate of S6K1, PDK1 could be phosphorylated at its C-terminal PH domain, which is critical for recruiting the adaptor protein 14-3-3 to interact with phosphorylated PDK1 species. Subsequently, 14-3-3 could dissociate PDK1 from PIP3 and plasma membrane, to promote PDK1 homo-dimerization, and further retard PDK1 interaction with and phosphorylation of the AKT kinase. More importantly, patient-associated *PDK1* mutations, either impairing S6K1-mediated phosphorylation or blocking 14-3-3 interaction, all display more malignant phenotypes than wild type PDK1 at least partially due to constitutively facilitating AKT kinase activity and oncogenic functions.

## Results

**S6K1 directly phosphorylates PDK1 at Ser549**. Serving as the unique kinase phosphorylating the T-loop (Thr308) of AKT, PDK1 plays a prominent role in full activation of the AKT kinase[6,7]. Although PDK1 is characterized to undergo auto-phosphorylation and activation, binding to PIP3 and localization on membrane are considered essential for PDK1 phosphorylating and activating the AKT kinase[6,7]. Thus, the post-translational modifications, in particular the phosphorylation event, occurred in PDK1 PH domain, is likely important for manipulating PDK1 function on governing AKT kinase activity. To this end, we screened the phosphorylation events in the PDK1 PH domain by a mass spectrometry (MS) approach coupled with data mining from the Cell Signaling Technology (CST) MS database (https://www.phosphosite.org), and several potential phosphorylation residues in the PDK1-PH domain were identified (Supplementary Fig. 1a–c). To further test the potential functions of these putative phosphorylation residues, we generated the phospho-deficient mutants for PDK1 (S/T to A), and observed that PDK1-S549A, to a lesser extent T522A, could markedly enhance PDK1 membrane translocation (Supplementary Fig. 1d). Furthermore, PI3K specific inhibitor BKM120 dramatically blocked both wild type (WT) and S549A PDK1 membrane translocation, indicating that the enhanced recruitment of PDK1-S549A to membrane is likely PI3K dependent (Supplementary Fig. 1e). Meanwhile, only the S549D mutation, but not other phospho-mimetic mutants of PDK1 including its auto-phosphorylation mutant (S241D), largely attenuated PDK1 binding to PIP3 (Supplementary Fig. 1f). In addition, PDK1-S549A promoted PDK1 colocalization with AKT in plasma membrane, while S549D displayed mainly plasma localization (Supplementary Fig. 1g). These findings together indicate the potent role of PDK1 phosphorylation at S549 in regulating its PIP3 binding and membrane localization.

To point out the potential upstream kinases mediated PDK1-S549 phosphorylation, we analyzed the surrounding amino acid of S549, and observed that the S549 site resided within a classic AGC kinase phosphorylation consensus motif (RxRxxS/T) (Fig. 1a). To assess whether the AGC family of kinases could promote PDK1 phosphorylation at this site, we screened a panel of AGC kinases, including the constitutively active AKT1 (myr-AKT1), AKT2 (myr-AKT2), S6K1 (S6K1-R3A) and SGK1 (SGK1-Δ60). Notably, we observed that S6K1, but not other AGC kinases we examined, markedly enhanced PDK1 phosphorylation detected with the pan-AKT substrate phosphorylation antibody at endogenous levels (Fig. 1b). Consistently, insulin-induced PDK1 phosphorylation was significantly antagonized by mTORC1 or S6K inhibitors, and to a lesser extent, AKT inhibitor, coupled with the decreased phosphorylation

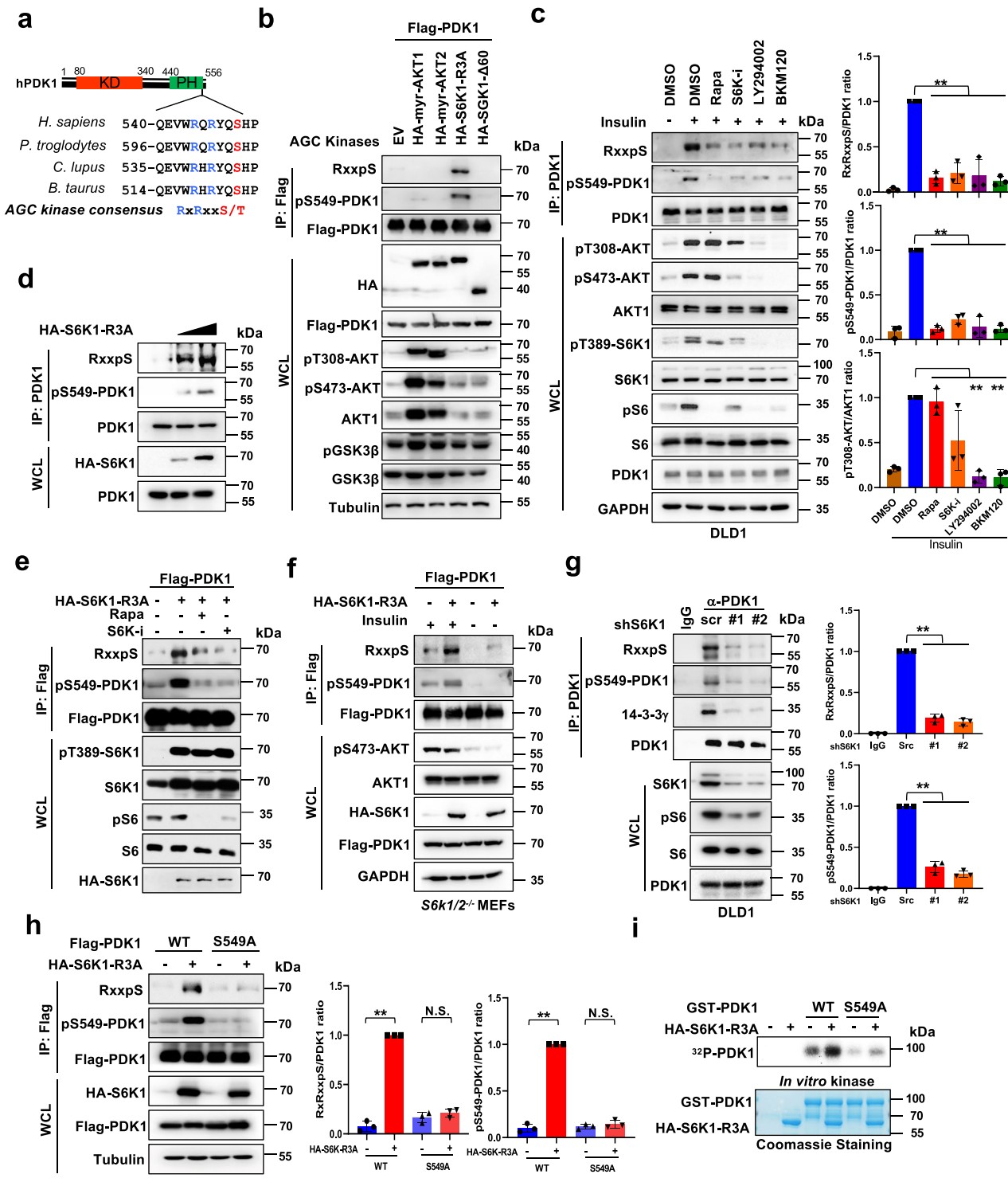

event of well-characterized S6K downstream substrate, ribosomal protein S6 (S6) (Fig. 1c and Supplementary Fig. 2a–e). In keeping with this notion, S6K1 could markedly enhance PDK1 phosphorylation in a dose dependent manner (Fig. 1d), while S6K1-induced PDK1 phosphorylation could be attenuated by mTORC1 or S6K inhibitor (Fig. 1e and Supplementary Fig. 2f). To further investigate the roles of insulin on stimulating PDK1 phosphorylation, we observed that PDK1 phosphorylation fluctuated upon insulin stimulation (Supplementary Fig. 2g). Moreover, insulin strongly accelerated PDK1 phosphorylation in S6K1 re-introduced $S6k1$ and $S6k2$ double knockout ($S6k1/2^{-/-}$)

mouse embryonic fibroblasts (MEFs) (Fig. 1f and Supplementary Fig. 2h), suggesting that S6K1 is likely essential for insulin-induced PDK1 phosphorylation. In keeping with this finding, depletion of $S6K1$ in cancer cells significantly reduced PDK1 phosphorylation (Fig. 1g).

To verify whether S549 is indeed the major S6K1 phosphorylation residue on PDK1, S549A substitution has reduced S6K1-mediated PDK1 phosphorylation (Fig. 1h). In keeping with this finding, WT but not the S549A mutant form of PDK1, could be directly phosphorylated by S6K1 in vitro (Fig. 1i). To further detect PDK1 phosphorylation at S549, we successfully generated

**Fig. 1 S6K directly phosphorylates PDK1 on Ser549. a** A schematic presentation of the evolutionarily conserved putative AGC kinase phosphorylation consensus in PDK1. **b** IB analysis of WCL and immunoprecipitates (IP) derived from HEK293T cells transfected with Flag-PDK1 and the indicated HA-tagged constitutive active AGC family kinases. **c** DLD1 cells were serum-starved for 24 h and then collected after treated with Rapamycin (20 nM), S6K1-I (10 μM), LY294002 (10 μM), BKM120 (10 μM) for 1 h before being stimulated with insulin (100 nM) for 30 min. The resulting cells were subjected for IP and IB analysis. Indicated protein were quantified. (mean ± SD, $n = 3$), **$P < 0.01$. **d** 293 T cells were transfected with increasing dose of HA-S6K-R3A and subjected for IP and IB analysis. **e** IB analysis of WCL and IP products derived from 293 T cells transfected with Flag-PDK1 and HA-S6K1-R3A, where indicated, cells were treated with Rapamycin (20 nM) and S6K1-I (10 μM) for 1 hr. **f** IB analysis of WCL and IP products derived from $S6k1/2^{-/-}$ MEFs transfected with indicated constructed, and were serum-starved for 24 h and then collected after insulin (100 nM) stimulation for 30 min. **g, h** IB analysis of IP products and WCL derived from DLD1 cells infected with shRNA against $S6K1$ **g** or 293 T cells transfected with indicated constructs. **h**. Indicated protein were quantified. (mean ± SD, $n = 3$, $P = 0.001, 0.435, 0.0009, 0.157$). **i** In vitro kinase assays were performed with recombinant bacterially purified PDK1 protein as substrates and IP S6K1 protein from 293 T cells was used as the source of kinase. $^{32}P$ isotope-ATP was used to detect the phosphorylated PDK1. Similar results were obtained in $n \geq 3$ independent experiments in **b, f, i**. Statistical significance was determined by two-tailed Student's $t$-test in **c, g, h**. N.S > 0.05, *$P < 0.05$,**$P < 0.01$. Source data are provided in Source Data files. EV, empty vector. WCL, whole cell lysate. IP, immunoprecipitation. Scr, scramble. WT, wild type.

and validated the phosphorylation antibody specifically recognizing PDK1-pS549 (Supplementary Fig. 3a–d), and observed that S6K1 indeed could phosphorylate PDK1 at S549 in cells (Fig. 1b–h), which could be inhibited by treating with mTORC1 or S6K inhibitor (Fig. 1c). To examine the phosphorylation event of PDK1-S549 under physiological conditions, we observed that insulin or EGF administration markedly enhanced the phosphorylation of WT compared to the S549A mutant form of PDK1 (Supplementary Fig. 3e–g). Hence, these findings together suggest that S6K1 directly phosphorylates PDK1 at the Ser549 residue.

**S6K1-mediated PDK1 phosphorylation inhibits AKT kinase activity and oncogenic functions**. To explore the biological function of S6K1-mediated PDK1 phosphorylation, we introduced different mutant forms of PDK1 (WT, S549A and S549D) into DLD1-$PDK1^{-/-}$ cells, and observed that, compared with WT or PDK1-S549D, PDK1-S549A expression significantly elevated AKT-pT308 (Fig. 2a). Furthermore, pT308-AKT and its downstream substrates, such as pGSK3β, could be significantly elevated in DLD1-$PDK1^{-/-}$ cells expressing PDK1-S549A compared with PDK1-WT- or S549D expressing cells under insulin treatment condition (Fig. 2b, c). Similar results were observed in 293 T cells depleted endogenous $PDK1$ by clustered regularly interspaced short palindromic repeats (CRISPR)-Cas9, followed by stably expressing EV, WT or S549A variant of PDK1 with the treatment of EGF or insulin, respectively (Supplementary Fig. 4a, b).

In keeping with the notion that AKT plays important oncogenic roles in governing cell proliferation, survival and tumorigenesis[9], we revealed that introduction of the phospho-deficient form of PDK1 (S549A) could significantly enhance cell colony formation and anchorage independent growth compared with WT or S549D mutant PDK1 (Fig. 2d, e). Meanwhile, PDK1-S549A expressing HEK293-sg$PDK1$ cells also displayed an augmented colony formation ability compared with WT-PDK1 expressing cells (Supplementary Fig. 4c). In keeping with the pivotal roles of AKT in anti-chemotherapy-induced apoptosis[22], PDK1-S549A expressing DLD1-$PDK1^{-/-}$ cells exhibited more resistance to the treatment of cisplatin or doxorubicin than WT or S549D-expressing DLD1-$PDK1^{-/-}$ cells (Supplementary Fig. 4d, e). Next, to reveal the in vivo functions of PDK1-S549 phosphorylation in tumor growth, we employed a xenograft mouse model, and observed that expression of PDK1-S549A could apparently facilitate DLD1-$PDK1^{-/-}$ cell tumor growth compared with PDK1 WT or S549D expressing DLD1-$PDK1^{-/-}$ cells (Fig. 2f–j). Meanwhile, tumors bearing PDK1-S549A exhibited more Ki67 staining coupled with increased AKT downstream substrate pS6, compared with the tumors bearing PDK1-WT or S549D expression (Fig. 2k, l and Supplementary

Fig. 4f). In addition, genetically depleting or pharmacologically inhibiting AKT1 could robustly attenuate PDK1-S549A-induced cancer cell colony formation (Supplementary Fig. 5a–c). These results together suggest that S6K-mediated PDK1 phosphorylation negatively regulates AKT kinase activity and oncogenic functions both in vitro and in vivo.

**PDK1-S549 phosphorylation markedly attenuates PDK1 interaction with PIP₃ and plasma membrane localization**. To investigate whether S6K1-mediated PDK1 phosphorylation affects its kinase activity, different mutant forms of PDK1 were subjected to PDK1 in vitro kinase assays. The results showed that co-transfection of the constitutively active form of S6K1-R3A with PDK1-WT, S549A or S549D variant only mildly affected PDK1 capability to phosphorylate its substrate AKT1 (Fig. 3a), indicating that the repression of AKT kinase activity by S6K1-mediated PDK1 phosphorylation is not likely due to reduced PDK1 enzymatic activity. Next, we monitored the interaction between PDK1 and AKT, and observed that S6K1-R3A could dramatically decrease the interaction of PDK1 with AKT in a dose-dependent manner coupled with increased PDK1 phosphorylation at S549 (Fig. 3b).

To exclude the influence of established negative feedback role of S6K by phosphorylating IRS1/2[23,24], we measured the interaction of PDK1-S549D or S549A with AKT, and observed that PDK1-S549A enhanced, whereas S549D attenuated, the interaction between PDK1 and AKT1 (Fig. 3c), or AKT2 and AKT3 (Supplementary Fig. 6a), but did not affect the interaction with other PDK1 substrates, such as SGK1, PLK1 or S6K1 (Supplementary Fig. 6c, d). Given that the interaction of PDK1 and AKT mainly occurred in the plasma membrane with PIP₃ localization[6], we also previously observed that the S549D mutation decreased the interaction of PDK1 with PIP₃ (Supplementary Fig. 1f), thus, we asked whether S6K1 led to decreased interaction of PDK1/AKT via impairing PDK1 interaction with PIP₃. To this end, PIP₃ pull-down assays were carried out, and the result showed that S6K1-R3A dramatically decreased the interaction of the WT, but not the S549A mutant form of PDK1 with PIP₃ (Fig. 3d). Meanwhile, pharmacologically inhibiting S6K1 activity could promote the interaction of PDK1 with PIP₃ (Fig. 3e). These data indicate that S549 phosphorylation could dissociate PDK1 from plasma membrane and PIP₃, resulting in decreased interaction of PDK1 with AKT. In keeping with this finding, we observed that S6K1 indeed could reduce PDK1 membrane localization by cell fractionation assays (Fig. 3g) or immune staining (Fig. 3h). As a result, compared with WT or S549D form of PDK1, the S549A mutant displayed an enhanced PDK1 membrane localization. Moreover, depletion of $S6K1$ not only enhanced endogenous PDK1 binding with PIP₃ (Supplementary Fig. 6e), but also promoted AKT1 and PDK1 co-translocate to plasma membrane (Supplementary Fig. 6f). Interestingly, enforced

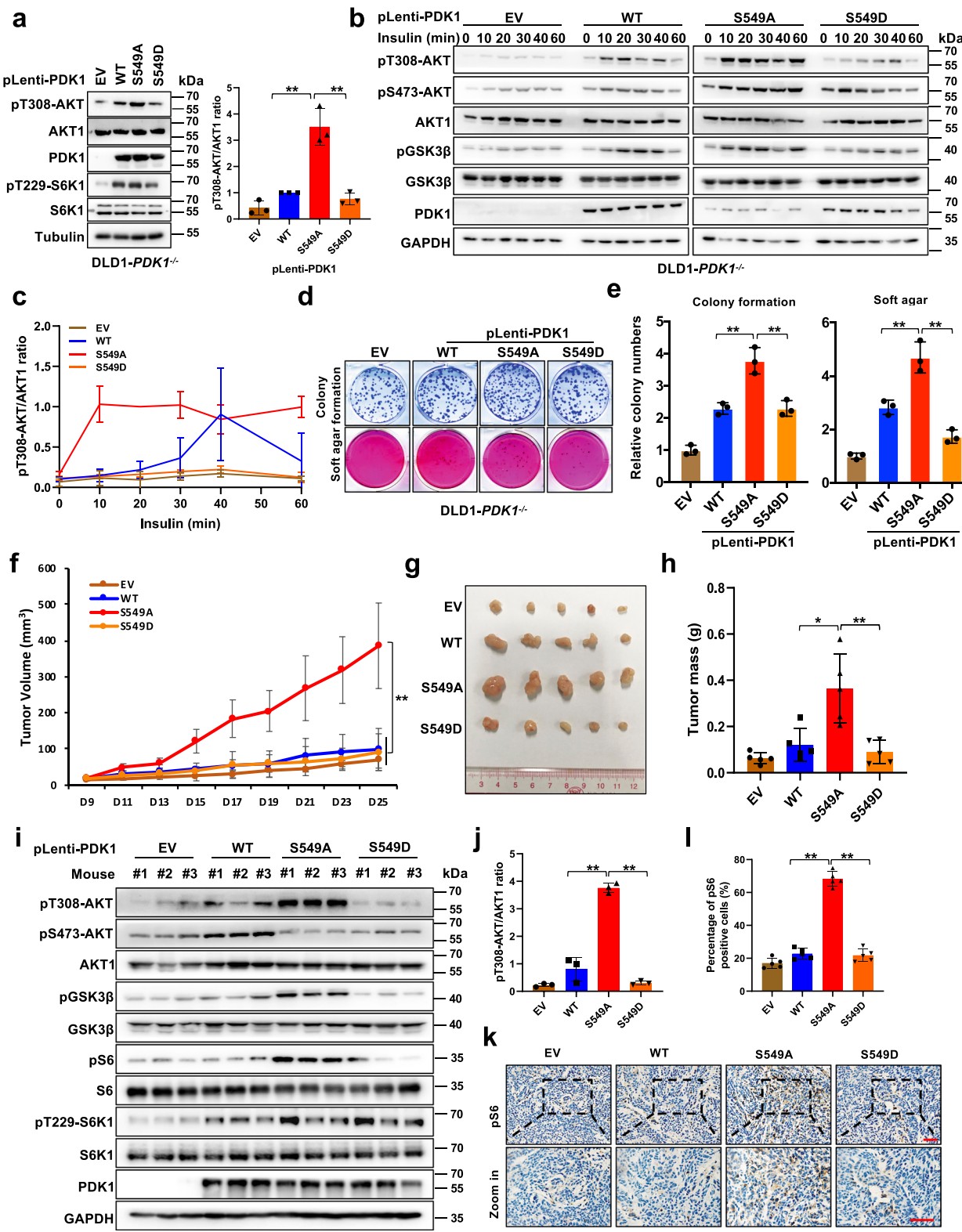

membrane location of PDK1 (Myr-PDK1) attenuated both PDK1-S549A-enhanced and PDK-S549D-decreased AKT activation (Supplementary Fig. 6g). These results indicate that S6K1 indeed affects PDK1/PIP₃ binding and membrane translocation likely in a phosphorylation dependent manner.

Homo-dimerization has been previously indicated as a negative regulatory mechanism of PDK1 on AKT kinase[25,26], however, the regulation of PDK1 dimerization is not uncovered yet. Hence, we

intended to evaluate whether S6K1-mediated PDK1 phosphorylation affected its homo-dimerization. To this end, we observed that PDK1 homo-dimerization was markedly enriched following ectopic expression of a dose-increased S6K1-R3A (Supplementary Fig. 6h), or phospho-mimetic mutant PDK1-S549D (Supplementary Fig. 6i). These results implicate that S6K1-mediated PDK1 phosphorylation enhances its dimerization and contributes to the negative regulation of AKT kinase.

**Fig. 2 Phosphorylation of PDK1 inhibits AKT kinase activity and oncogenic functions. a** IB analysis of DLD1-*PDK1* knockout cells stably infected with the indicated constructs. Indicated protein were quantified. (mean ± SD, $n = 3$, $P = 0.004$, 0.003). **b** IB analysis of WCL derived from DLD1-*PDK1* knockout cells transfected with the indicated Flag-PDK1 constructs that were serum-starved for 12 h and then treated with the insulin (100 nM) for the indicated time periods before collection for IB analysis. **c** pT308-AKT/AKT1 ratio in **b** were calculated. **d, e** Cells generated in **a** were subjected to colony formation and soft agar assays. (mean ± SD, $n = 3$, $P = 0.003$, 0.003, 0.005, 0.002). **f, g, h** Cells generated in **a** were subjected to mouse xenograft assays. Tumor sizes were monitored (**f** $P = 0.0002$), and dissected tumors were weighed and calculated (**g, h**, $P = 0.0042$, 0.0059). The Error bars in the **f, h** are mean plus or minus SD, $n = 5$ mice. *$P < 0.05$, **$P < 0.01$. **i** IB analysis of WCL derived from dissected tumor tissues. **j** pT308-AKT/AKT1 ratio in (**i**) were calculated. (mean ± SD, $n = 3$, $P = 0.0033$, 0.0018), **k, l** Graphic representation of pS6 staining derived from tumor tissues **k**, which were further normalized and quantified **l**. (mean ± SD, $n = 5$, $P = 6.41E-06$, 2.29E-05) (*t* test), scale bar, 50 μm. Similar results were obtained in $n \geq 3$ independent experiments in **b**. Statistical significance was determined by two-tailed Student's *t*-test in **a, e, h. j, l** and by two-way ANOVA in **f**. *$P < 0.05$, **$P < 0.01$. Source data are provided in Source Data files. EV, empty vector. WT, wild type.

## 14-3-3 mediates dissociation of PDK1 from PIP₃ and enhances its dimerization

To uncover the underlying mechanism of phosphorylated PDK1 disassociation from $PIP_3$ and forming dimerization, we hypothesized that there would be adaptor proteins recognizing phosphorylated PDK1 and further disturb its interaction with $PIP_3$. It is well-established that 14-3-3 adaptor proteins play primary roles in protein compartment trans-localization, especially for the phosphorylated proteins[27,28]. Thus, we screened the 14-3-3 family members and found that 14-3-3γ could specifically interact with PDK1 in both exogenous and endogenous levels (Fig. 4a–c). Since 14-3-3 prefers to interact with the phosphorylated motif (RxxpSxP), which has been previously observed in the PH domain of PDK1 (Fig. 1a). Of note, this motif overlapped with S6K1-mediated PDK1 phosphorylation residue (Ser549), and evolutionarily conserved among different species (Fig. 4d). Interestingly, compared with different PH domains derived from PDK1, AKT1, PLK1, and SIN1, we observed that only PDK1-PH engaged in binding with 14-3-3γ (Supplementary Fig. 7a). More importantly, S6K1-R3A markedly enhanced, while depletion of *S6K1* decreased the interaction of PDK1 with 14-3-3γ (Fig. 1g and Supplementary Fig. 7b). In keeping with these findings, S549D mutant PDK1 enhanced, whereas S549A mutant PDK1 attenuated the interaction of PDK1 with 14-3-3γ both in cells and in vitro (Fig. 4e, f and Supplementary Fig. 7c).

It is previously reported that the proline residue followed the phosphorylated serine or threonine is critical for 14-3-3 interaction[28]. As a result, the mutation of PDK1-P551A strongly attenuated the interaction of PDK1 with 14-3-3γ both in cells or in vitro (Supplementary Fig. 7c, d). Interestingly, the P551A mutant did not abrogate S6K-mediated dissociation of PDK1 with $PIP_3$ (Fig. 4g), resulting in the increase of its interaction with AKT (Supplementary Fig. 7e), which could not be compromised by ectopically expressing S6K1-R3A (Fig. 4g). Consistently, ectopic expression of S6K1-R3A or PDK1-S549D enhanced the recruitment of 14-3-3γ to PDK1 (Fig. 4g and S7f). By contrast, the P551A mutant efficiently disrupted the interaction of PDK1 with 14-3-3γ (Supplementary Fig. 7f), and enhanced its binding with $PIP_3$, which could not be disturbed by expressing S6K1-R3A (Fig. 4g and Supplementary Fig. 7g). This result indicates that S6K1-mediated PDK1 phosphorylation is necessary for recruiting 14-3-3γ, which is the major route responsible for phosphorylation-mediated dissociation of PDK1 from $PIP_3$. To further confirm the pivotal role of 14-3-3γ in regulation of PDK/$PIP_3$ interaction, concurrent mutations of S549D and P551A (termed PDK1-S549D-P551A) were generated and markedly impaired the interaction of 14-3-3γ/PDK1 (Supplementary Fig. 7f), instead of efficiently promoting the binding of PDK1/$PIP_3$ (Supplementary Fig. 7g), and PDK1/AKT (Supplementary Fig. 7h), respectively. Consistent with the phenotype that PDK1-S549A enhanced AKT phosphorylation at T308, PDK1-P551A also dramatically enhanced AKT-pT308 upon insulin treatment (Fig. 4i).

In further supporting the potent roles of 14-3-3γ in regulating PDK1 dimerization, via a gel-filtration approach, we observed that PDK1 and pS549-PDK1 were colocalized in the similar fraction with 14-3-3γ at its dimer size around 150KD (Fig. 4j), indicating that 14-3-3γ possibly assembles the phosphorylated PDK1 for dimerization. Meanwhile, depletion of endogenous *14-3-3γ* significantly enhanced PDK1/$PIP_3$ interaction (Supplementary Fig. 7i), and impaired S6K1-mediated disassociation of PDK1 from $PIP_3$ (Supplementary Fig. 7j). Furthermore, depleting *14-3-3γ* abrogated PDK1 dimerization (Fig. 4k), and increased PDK1 interaction with AKT (Supplementary Fig. 7k). More importantly, knockdown of *14-3-3γ* markedly elevated the phosphorylation of AKT and its downstream target (Fig. 4l, m), indicating that the adaptor protein 14-3-3γ plays an important role in S6K1-mediated AKT repression by PDK1. These findings together implicate a model that 14-3-3γ binds to PDK1 in an S6K1-mediated S549 phosphorylation dependent manner, which dissociates PDK1 from $PIP_3$, in turn promotes PDK1 dimerization, resulting in repression of AKT kinase activity (Fig. 4n).

## Patient-associated mutations of *PDK1* impair its interaction with 14-3-3 and facilitate AKT kinase activity and oncogenic functions

To investigate the pathological roles of S6K1-mediated PDK1 phosphorylation in tumorigenesis, we analyzed the cBio-Portal of Cancer Genomics database (www.cbioportal.com)[29], and observed that patients-associated *PDK1* mutations occurred around S6K1-mediated PDK1 phosphorylation or 14-3-3γ binding regions, although only once in distinct cohort was observed (Fig. 5a). Among these mutations, P551A and P551Q mutations were observed once from thyroid cancer (total 496) and melanoma (total 359) respectively in the TCGA cohorts[29], S549N was observed once from the DFCI colorectal adenocarcinoma cohorts (total 619)[30], and R544K and R546P were observed once from lung (total 604) and colon (total 138) adenocarcinoma respectively in the MSK-IMPACT cohorts[31]. Whereas the P551L mutation was observed from an oral epithelial cancer cell line in the Cancer Cell Line Encyclopedia (total 1739)[32]. Next, these mutations were divided into two groups, type I mutations (R544K, R546P and S549N) were termed due to their location in the region of possibly blocking S6K1-mediated phosphorylation; type II mutations (P551A, P551L and P551Q) were termed due to their possible role in directly disturbing the re-cognization of 14-3-3γ (Fig. 5a). Of note, type I, a lesser extent of type II mutants, markedly decreased S6K1-mediated PDK1 phosphorylation (Fig. 5b and Supplementary Fig. 8a, b). Whereas, both type I and II mutants dramatically attenuated PDK1/14-3-3γ interaction (Fig. 5c and Supplementary Fig. 8c), resulting in increased PDK1 binding to $PIP_3$ (Fig. 5d), and their membrane localization (Fig. 5e–h). Notably, these mutations enhanced the interaction of PDK1 with AKT (Fig. 5i, j), coupled with increased phosphorylation of AKT and its targets in HEK293-sg*PDK1* cells (Fig. 5k, l).

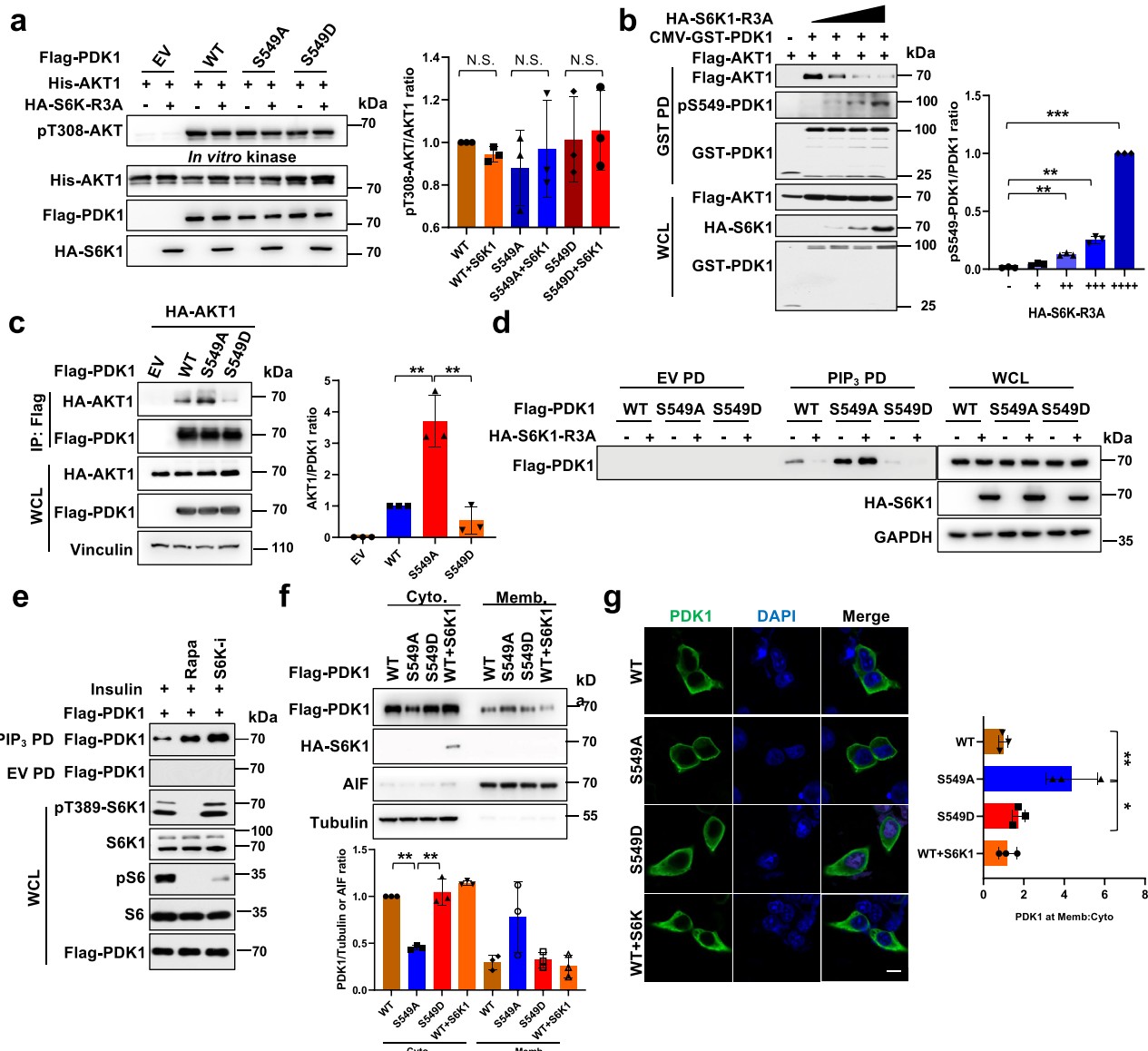

**Fig. 3 S6K1-mediated phosphorylation of PDK1 inhibits PDK1 membrane localization and PIP3 binding. a** The indicated Flag-PDK1 constructs with/without HA-S6K1-R3A were transfected into 293 T cells and anti-Flag immunoprecipitation was recovered as the kinase source to phosphorylate insect cells purified His-AKT1 in vitro. pT308-AKT/AKT1 ratio were calculated. (mean ± SD, n = 3). **b** IB analysis of WCL and GST-pulldown derived from 293 T cells transfected with GST-PDK1, Flag-AKT1 and increasing dose of HA-S6K1-R3A. pS549-PDK1/PDK1 ration were calculated. (mean ± SD, n = 3, P = 0.0001, 0.0002, 7.82E-10). **c** IB analysis of WCL and IP products derived from 293 T cells transfected with Flag-PDK1(WT, S549A, S549D) and HA-AKT1. Indicated protein were quantified. (mean ± SD, n = 3, P = 0.005, 0.004). **d** IB analysis of PIP₃ pull-down products and WCL derived from 293 T cells transfected with the indicated constructs. Where indicated, empty beads (EV) serve as a negative control. **e** IB analysis of PIP₃ pull-down products and WCL derived from HeLa cells transfected with Flag-PDK1 that were serum-starved for 24 h and then collected after insulin (100 nM) stimulation for 30 min, where indicated, the kinase inhibitors were added. **f** IB analysis of cell fractionations separated from 293 T cells transfected with indicated constructs. PDK1/Tubulin or AIF ratio were calculated. (mean ± SD, n = 3, P = 2.98E-06, 0.002). **g** Representative immunofluorescence images of 293 T cells transfected with indicated constructs, scale bar, 10 μm. Mean PDK1 fluorescence intensity at plasma membrane relative cytosol was determined, data represent mean ± SD, P = 0.008, 0.03. Greater than 60 cells were analyzed from 3 independent experiments. Similar results were obtained in n ≥ 3 independent experiments in **d**, **e**. Statistical significance was determined by two-tailed Student's t-test in **b**, **c**, **f**. **g** N.S > 0.05, *P < 0.05,**P < 0.01. Source data are provided in Source Data files. EV, empty vector. WCL, whole cell lysate. IP, immunoprecipitation. WT, wild type. PD, pulldown. Cyto, cytoplasm. Memb, membrane.

Furthermore, re-introducing these variants into DLD1-*PDK1*⁻/⁻ or SW480-sh*PDK1* cells increased pT308-AKT (Fig. 6a and Supplementary Fig. 8d), and robustly promoted cancer cell oncogenic capabilities measured by colony formation and soft agar assays (Fig. 6b, c and Supplementary Fig. 8e). Additionally, compared with PDK1-WT, ectopically expressing these variants markedly enhanced drug resistance of DLD1 cells to cisplatin and

doxorubicin (Supplementary Fig. 8f, g). Moreover, these mutants also facilitated tumor growth in the xenograft mouse models compared with WT-PDK1 expressing cells, coupled with increased AKT downstream pS6, as well as the proliferation marker Ki67 (Fig. 6d–g and Supplementary Fig. 8h–l). To point out whether the oncogenic roles of patient derived *PDK1* mutations due to elevating the AKT activity, we depleted *AKT1*

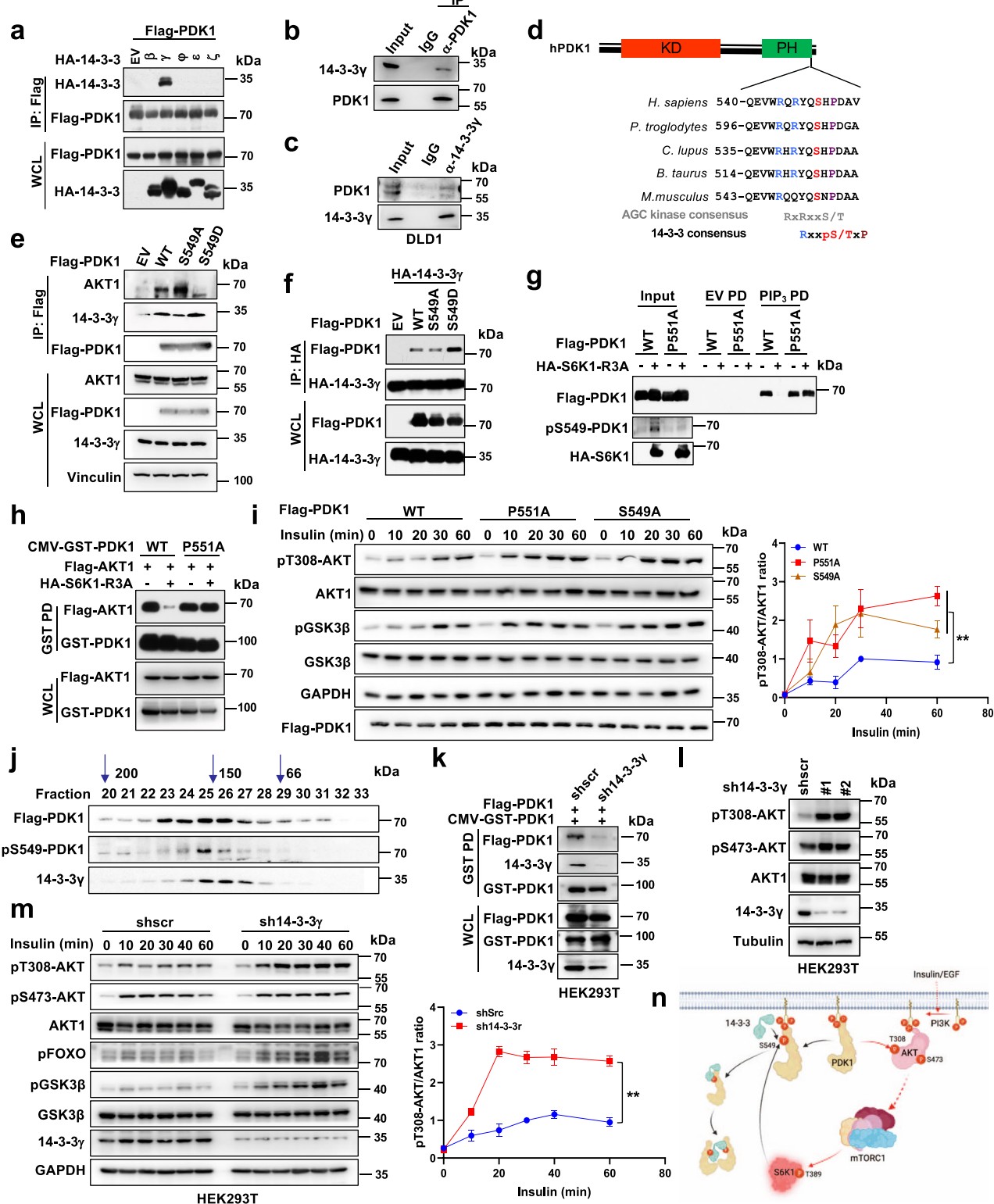

in either Type I or Type II mutant *PDK1*-expressing colon cancer cells (Supplementary Fig. 9a), and observed that knockdown *AKT1* markedly compromised the oncogenic roles of mutant *PDK1* in promoting colony formation (Supplementary Fig. 9b, c). These observations together indicate that pathological mutations derived from the PDK1-PH domain possibly impair S6K1-mediated PDK1 phosphorylation and 14-3-3 binding, thereby facilitating AKT kinase activity and oncogenic functions (Fig. 7).

## Discussion

Denoted as a central module of oncogenic signaling pathway, PI3K/AKT axis physiologically plays critical roles in manipulating cell proliferation, survival and metabolic homeostasis (Fig. 7a), and pathologically contributes to tumorigenesis and diabetes[9]. The regulation of AKT kinase has been extensively investigated, especially in the level of post-translational modification, including but not limited to phosphorylation, ubiquitination, hydroxylation

**Fig. 4 14-3-3 binds phosphorylated PDK1 and dissociates it from the cytoplasm membrane. a** IB analysis of WCL and IP products derived from 293 T cells transfected with Flag-PDK1 and the indicated various HA-tagged 14-3-3 constructs. **b, c** DLD1 cell lysates were subjected to IP with control IgG, anti-PDK1 or 14-3-3γ antibodies for IB analysis. **d** A schematic presentation of the evolutionarily conserved putative AGC kinase phosphorylation consensus in PDK1 and 14-3-3 recognized consensus. **e** IB analysis of WCL and IP products derived from 293 T cells transfected with indicated constructs. **f** IB analysis of WCL and IP products derived from 293 T cells transfected with HA-14-3-3γ and indicated constructs. **g** IB analysis of PIP$_3$ pull-down products and WCL derived from 293 T cells transfected with the indicated constructs. **h** IB analysis of WCL and GST-pulldown derived from 293 T cells transfected with the indicated constructs. **i** IB analysis of WCL derived from 293 T cells transfected with the indicated Flag-PDK1 constructs that were serum-starved for 12 h and then treated with insulin (100 nM) for the indicated time periods before collection for IB analysis. pT308-AKT/AKT1 ratio were calculated. (mean ± SD, n = 3, P = 0.0003). **j** IB analysis of the indicated fractionations derived from the gel filtration experiment with 293 T cells transfected with Flag-PDK1. **k** IB analysis of WCL and GST-pull-down derived from 293 T control or *14-3-3γ* knockdown cells transfected with GST-PDK1 and Flag-PDK1. **l** IB analysis of WCL derived from 293 T control or *14-3-3γ* knockdown cells. **m** IB analysis of WCL derived from 293 T control or *14-3-3γ* knockdown cells that were serum-starved for 12 h and then treated with insulin (100 nM) for the indicated time periods before collection for IB analysis. pT308-AKT/AKT1 ratio were calculated. (mean ± SD, n = 3, P = 0.0001). **n** Proposed model for 14-3-3 involved in the PDK1/AKT pathway regulation. Red arrows indicate positive regulation, and blue arrows indicate negative regulation. Similar results were obtained in n ≥ 3 independent experiments in **a, b, c, e, f, g, h**. Statistical significance was determined by two-way ANOVA in **i, m**. **\*\*P < 0.01. Source data are provided in Source Data files. EV, empty vector. WCL, whole cell lysate. IP, immunoprecipitation. WT, wild type. PD, pulldown. Scr, scramble.

and methylation[33,34], however, the direct regulation on its critical upstream kinase PDK1 remains largely mystic. Meanwhile, recently multiple negative feedback pathways governing AKT kinase activity have been evaluated, including S6K1-mediated phosphorylation of insulin receptor substrate 1 (IRS-1) to modulate PI3K[23,24], S6K1-mediated phosphorylation of Sin1 to impair mTORC2 complex[35], and mTORC1-mediated growth factor receptor bound protein 10 (Grb10) phosphorylation to disturb PI3K[36,37] (Fig. 7b). However, whether there are direct modifications on PDK1 to feedback regulating AKT kinase is still largely unclear. Here, we report another layer of negative feedback regulation on the AKT kinase, which is also mediated by the S6K1 kinase, via directly phosphorylating PDK1 on its PH domain. In short, S6K1 directly phosphorylates PDK1 PH domain to enhance PDK1 binding 14-3-3, which further dissociates PDK1 from PIP$_3$ and plasma membrane, resulting in the decrease of PDK1/AKT interaction and impairment of AKT-T308 phosphorylation. This observation suggests that S6K1 tightly controls AKT kinase activity in distinct negative feedback fashions and phosphorylation dependent manners (Fig. 7c).

The biological functions of adaptor protein 14-3-3 have been well-linked to many important pathophysiological signaling processes via its feature to interact with the Ser/Thr phosphorylated proteins, leading cellular compartment translocation[27,28]. Although the function of 14-3-3 proteins have been reported to negatively regulate the PDK1/AKT signal pathway[26], the underlying mechanism is not well defined to date. In our work, the 14-3-3γ protein has been integrated into the regulation of PDK1, not only by dissociating PDK1 from PIP$_3$, but also by enhancing the formation of PDK1 homo-dimerization in an S6K1-mediated phosphorylation dependent manner (Fig. 4j and S6g–h). To validate the biological roles of 14-3-3γ in regulating PDK1/AKT signaling, we deplete endogenous *14-3-3γ* in cells and reveal the potent role of 14-3-3γ in inhibiting AKT kinase by manipulating PDK1 localization and dimerization, thus promoting cancer cell malignant phenotypes. Interestingly, 14-3-3γ has also been considered to play an important role in negatively regulating the programmed death ligand 1 (PD-L1) abundance by manipulating SPOP protein to influence immune response for cancer therapies[38]. Therefore, perturbation of 14-3-3γ not only facilitates tumor evading immune-surveillance by enhancing PD-L1 levels, but also benefits cancer cell proliferation and anti-apoptosis capability by activating the AKT kinase.

Mutations frequently occurred in the components of PI3K/AKT signaling pathway, such as the well-established loss-of-function mutations of *PTEN*[39] and *von Hippel-Lindau tumor suppressor (VHL)*[40] or gain-of-function mutations of *PIK3CA*[41], *KRAS*[42] or *AKT*[43], all contribute to AKT kinase activity and cancer cell

malignancies. Although *PDK1* has been shown accumulated in many tumors and contributes to tumorigenesis by activating AKT kinase, its somatic mutations in tumorigenesis are rarely investigated yet. Although we identify cancer patient-associated gain-of-function mutations in the *PDK1* PH domain with a very rare (<1% of cases) compared with other components of PI3K-AKT pathway, these mutations either disturb S6K1-mediated PDK1 phosphorylation or block 14-3-3 binding the phosphorylated PDK1, also resulting in unleashing the inhibitory roles of S6K1 on PDK1/AKT signaling to exhibit oncogenic functions (Fig. 7c). To further verify the potent roles of these patient-derived *PDK1* mutations, knock in mouse models will be employed in future to monitor the mouse body size changes and their sensitivity to tumorigenesis similar as we have done previously[44]. In summary, our findings not only reveal an important feedback regulation of S6K on PDK1/AKT signaling pathway by directly phosphorylating PDK1, but also provide a potential strategy to combat gain-of-function mutant *PDK1*-driven cancers.

## Methods

**Cell culture, transfection and cell fractionations**. HEK293, 293 T and DLD1 cells were obtained from American Type Culture Collection (ATCC), DLD1-*PDK1$^{-/-}$* and counterpart cells were kindly provided by Dr. Bert Vogelstein (Johns Hopkins University School of Medicine), and these cells were cultured in Dulbecco's modified Eagle's medium (DMEM) supplemented with 10% FBS. Cell transfection was performed using Lipofectamine and Plus reagents, as described previously[44]. Packaging of lentiviral shRNA or cDNA expressing viruses and retroviral cDNA expressing viruses, as well as subsequent infection of various cell lines were performed according to the protocols described previously[44]. Following viral infection, cells were maintained in the presence of puromycin (0.5 μg/ml).

Cell fractionations were performed with Cell Fractionation Kit (CST9038). Kinase inhibitors Mk2206 (Selleck S1078), Rapamycin (Selleck S1039), PP242 (Selleck S2218) and PF-4708671 (Selleck S2163) were used at the indicated doses. Growth factors including EGF (Sigma E9644) and insulin (Invitrogen 41400-045), were used at the indicated doses. PIP$_3$ beads (P-B00Ss) and label-free PIP$_3$ (P-3908) were purchased from Echelon Biosciences.

**Plasmid construction**. Constructs of pcDNA3-HA-S6K1, pcDNA3-HA-AKT1, pcDNA3-HA-AKT2, pcDNA3-HA-AKT3 and pcDNA3-HA-myr-AKT1, pcDNA3-HA-myr-AKT2, pcDNA3-HA-S6K1-R3A, pcDNA3-HA-Δ60-SGK1, pCMV-Flag-PDK1, pCMV-Flag-AKT1 were previously described[35]. pCMV-GST-PDK1, pCMV-GST-14-3-3γ, pCMV-GST-Akt1 were cloned into mammalian expression GST-fusion vectors. pcDNA3-HA-SGK1, pcDNA3-HA-14-3-3β, pcDNA3-HA-14-3-3γ, pcDNA3-HA-14-3-3φ, pcDNA3-HA-14-3-3ε, pcDNA3-HA-14-3-3ζ were cloned into mammalian expression HA-fusion vectors. pcDNA3-Myc-PDK1, pcDNA3-Myc-PLK1 and pcDNA3-Myc-myr-PDK1 were cloned into mammalian expression Myc-fusion vectors. pLenti-puro-PDK1 was cloned into mammalian expression pLenti-puro vectors. Details of plasmid constructions are available upon request.

Various PDK1 mutants were generated using the QuikChange XL Site-Directed Mutagenesis Kit (Stratagene) according to the manufacturer's instruction. All mutants were generated using mutagenesis PCR and the sequences were verified by DNA sequencing.

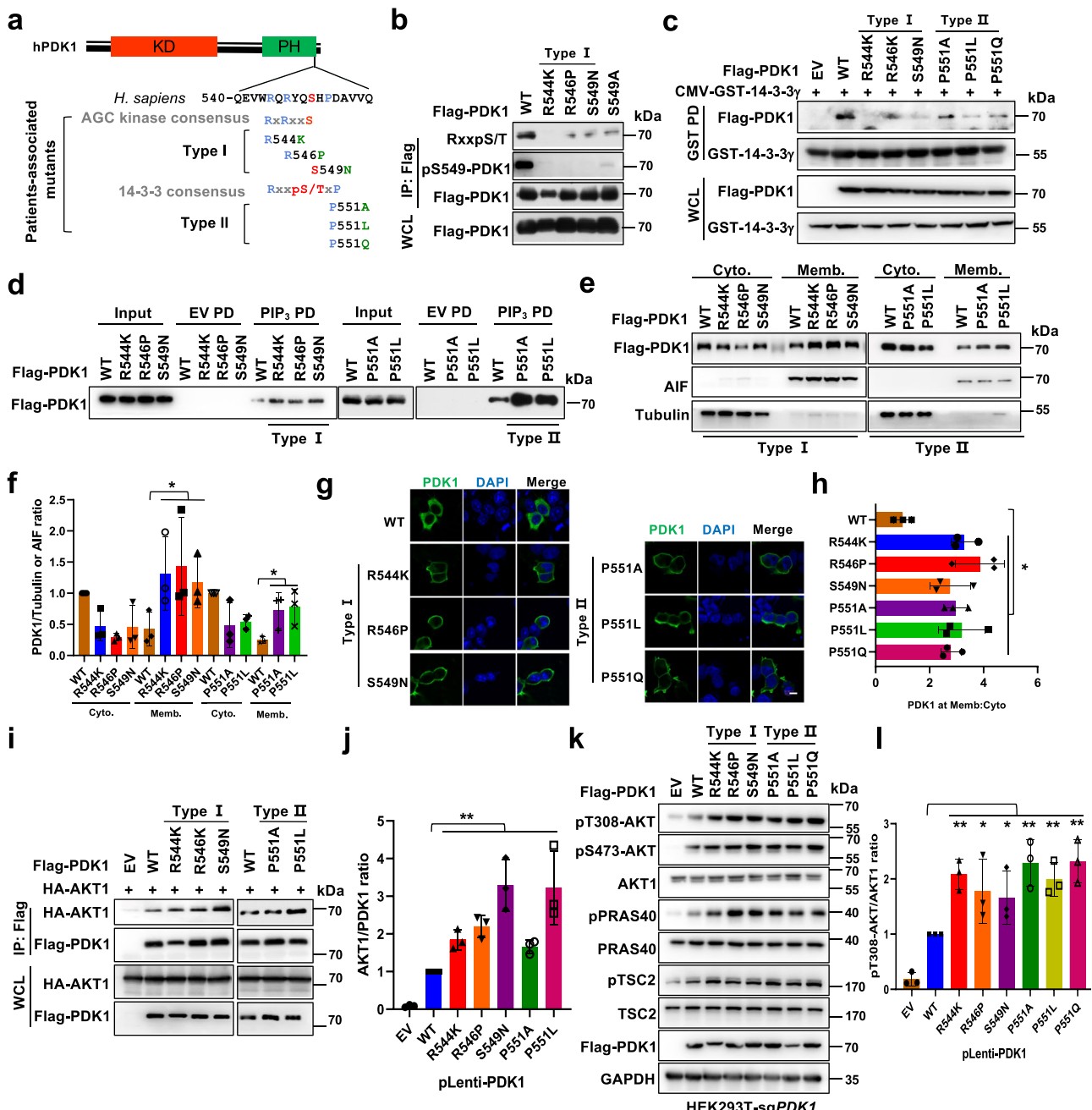

**Fig. 5 Patients-associated mutations of PDK1 confer to PDK1 membrane location and AKT kinase activation. a** A schematic presentation of the patients-associated *PDK1* mutations occurred around S6K1-mediated PDK1 phosphorylation or 14-3-3γ binding region. **b** IB analysis of WCL and IP products derived from 293 T cells transfected with the indicated constructs. **c** IB analysis of WCL and GST-pull-down derived from 293 T cells transfected with GST-14-3-3γ and the indicated constructs. **d** IB analysis of PIP₃ pull-down products and WCL derived from 293 T cells transfected with Flag-PDK1(WT, R544K, R546P, S549N, P551A, P551L). **e** IB analysis of cell fractionations separated from 293 T cells transfected with the indicated constructs. **f** PDK1/Tubulin or AIF ratio in **e** were calculated, (mean ± SD, *n* = 3), *\**P* < 0.05. **g** Representative immunofluorescence images of 293 T cells transfected with the indicated constructs, scale bar, 10 μm. **h** Mean PDK1 fluorescence intensity at plasma membrane relative cytosol was determined, data represent mean ± SD, *P* = 0.022, 0.036, 0.011, 0.035, 0.026, 0.030. Greater than 60 cells were analyzed from 3 independent experiments. **i** IB analysis of WCL and IP products derived from 293 T cells transfected with HA-AKT1 and the indicated construct. **j** AKT1/PDK1 ratio in **i** were calculated, (mean ± SD, *n* = 3), *\**P* < 0.05. **k** IB analysis of WCL derived from 293T-*PDK1* knockout cells transfected with the indicated constructs. **l** pT308-AKT/AKT1 ratio in **k** were calculated, (mean ± SD, *n* = 3), *\**P* < 0.05, *\*\**P* < 0.01. Similar results were obtained in *n* ≥ 3 independent experiments in **b**, **c**, **d**. Statistical significance was determined by two-tailed Student's *t*-test in **f**, **j**, **l**. *\**P* < 0.05, *\*\**P* < 0.01. Source data are provided in Source Data files. EV, empty vector. WCL, whole cell lysate. IP, immunoprecipitation. WT, wild type. PD, pulldown. Cyto, cytoplasm. Memb, membrane.

**Antibodies**. All antibodies were diluted in TBST buffer with 5% non-fat milk for western blot. Anti-AKT Substrate (RxxpS/T) antibody (9614, 1:1000), anti-phospho-Ser473-AKT antibody (4060, 1:3000), anti-phospho-Thr308-AKT antibody (2965, 1:1000), anti-AKT total antibody (4691, 1:3000), anti-PDK1 antibody (13037, 1:1000), anti-phospho-Ser241-PDK1 antibody (3061, 1:3000), anti-AIF antibody

(5318, 1:1000), anti-phospho-Ser9-GSK3β antibody (5558, 1:3000), anti-GSK3β antibody (12456, 1:1000), anti-phospho-FOXO1 (Ser256) antibody (9461, 1:1000), anti-Myc antibody (2276, 1:1000), anti-GST antibody (2625, 1:1000), anti-pS6K1 (Thr389) antibody (9205, 1:1000), anti-S6K1 antibody (2708, 1:1000), anti-S6 antibody (2217, 1:13000) and anti-pS240/244-S6 antibody (5364, 1:3000) were obtained

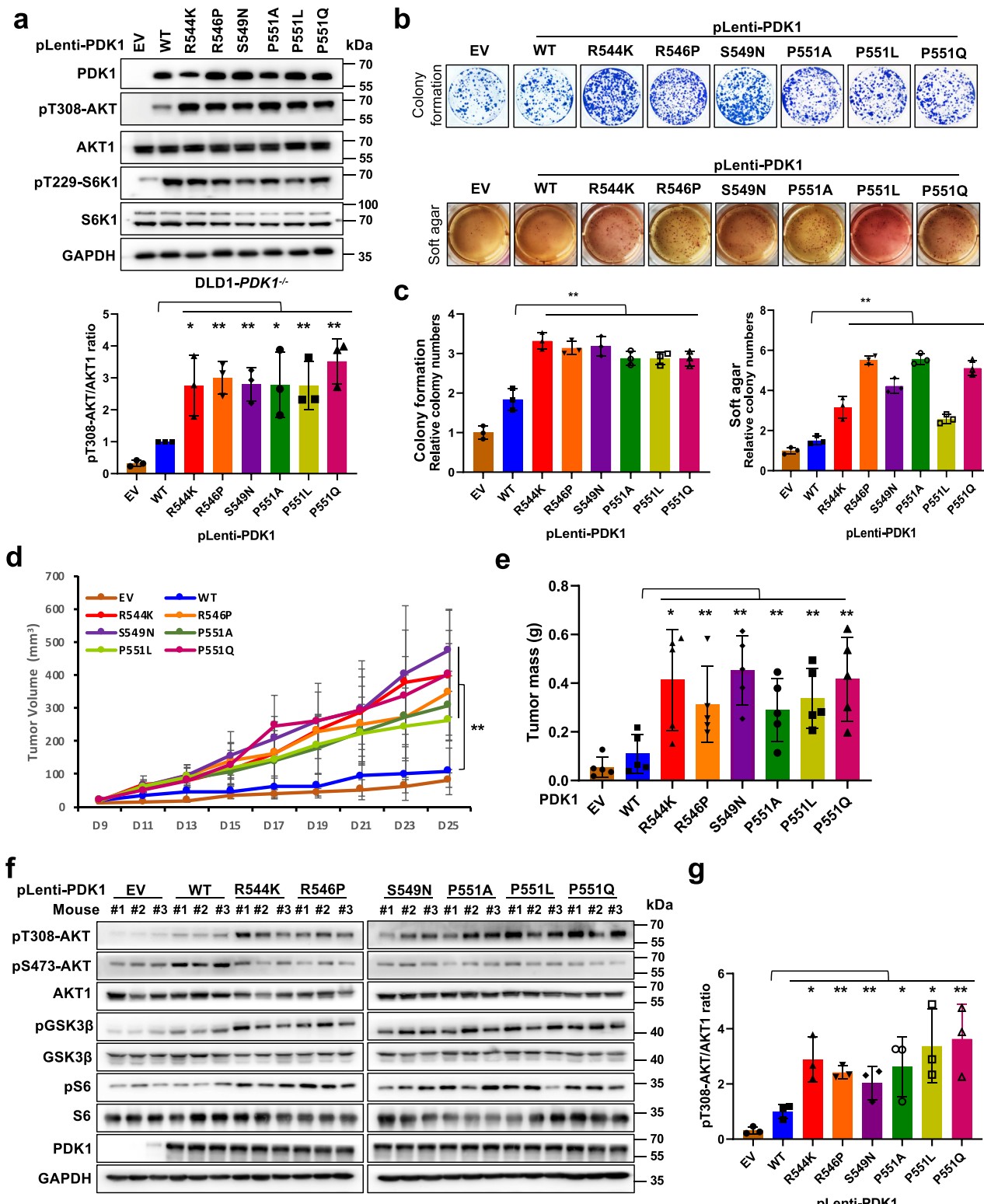

**Fig. 6 Patient-derived mutations of *PDK1* promote AKT1 activation and oncogenic functions. a** IB analysis of DLD1-*PDK1* knockout cells stably infected with the indicated constructs. pT308-AKT/AKT1 ratio were calculated. (mean ± SD, *n* = 3), *\*P* < 0.05, *\*\*P* < 0.01. **b**, **c** Cells generated in **a** were subjected to colony formation and soft agar assays. (mean ± SD, *n* = 3) *\*\*P* < 0.01. **d**, **e** Cells generated in **a** were subjected to mouse xenograft assays. Tumor sizes were monitored (**d**, *P* = 0.0045), and dissected tumors were weighed (**e**, *P* = 0.013, 0.008, 0.002, 0.003, 0.0003, 0.003). The Error bars in the **d**, **e** are mean plus or minus SD, *n* = 5 mice. *\*P* < 0.05, *\*\*P* < 0.01. **f** IB analysis of WCL derived from dissected tumor tissues. **g** pT308-AKT/AKT1 ratio in **f** were calculated, (mean ± SD, *n* = 3), *\*P* < 0.05, *\*\*P* < 0.01. Statistical significance was determined by two-tailed Student's *t*-test in **a**, **c**, **e**, **g** and two-way ANOVA in **d**. *\*P* < 0.05, *\*\*P* < 0.01. Source data are provided in Source Data files. EV, empty vector. WCL, whole cell lysate. WT, wild type.

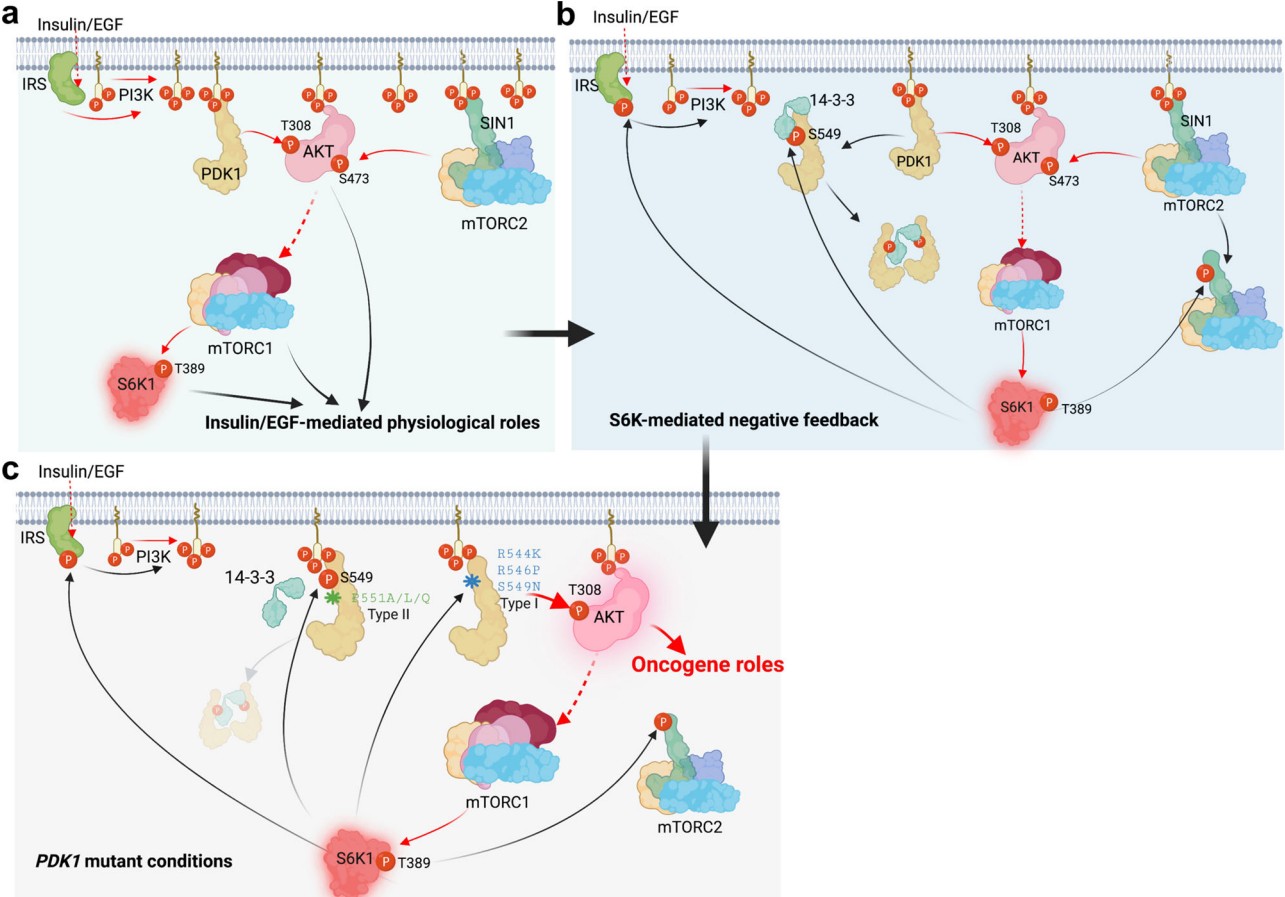

**Fig. 7 Models depicting the negative regulation of PDK1 by S6K1 under physiological and pathological conditions. a** The canonical activation of PI3K-AKT-mTOR pathway toward growth factors, such as Insulin or EGF treatment. **b** The negative feedback regulations of AKT kinase by S6K1-mediated phosphorylation of SIN1, PDK1 as well as IRS1. Among which, phosphorylation of IRS1 decreases PI3K-mediated PIP$_3$ generation; phosphorylation of SIN1 dissociates mTORC2 from membrane location; phosphorylation of PDK1 recruits 14-3-3 and dissociates from membrane location. These pathways together tightly negatively control growth factors-induced constant activation of AKT kinase. **c** Patient-associated *PDK1* mutations could block S6K-mediated PDK1 phosphorylation (Type I mutations) or 14-3-3-mediated PDK1 membrane dissociation (Type II mutations), leading to AKT constitutive activation and oncogenic roles, such as accelerating cell proliferation and anti-apoptosis. Red and black arrows indicate positive and negative regulation, respectively. Solid and dotted lines indicate direct and indirect (multistep) regulation, respectively.

from Cell Signaling Technology. Anti-pS6K1 (Thr229) antibody (ab5231, 1:1000) was obtained from Abcam. Polyclonal anti-HA antibody (sc-805, 1:1000) was obtained from Santa Cruz. Polyclonal anti-Flag antibody (F-2425, 1:1000), monoclonal anti-Flag antibody (F-3165, clone M2, 1:3000), anti-Tubulin antibody (T-5168, 1:3000), anti-Vinculin antibody (V4505, 1:3000), anti-Flag agarose beads (A-2220), anti-HA agarose beads (A-2095), peroxidase-conjugated anti-mouse secondary antibody (A-4416) and peroxidase-conjugated anti-rabbit secondary antibody (A-4914) were obtained from Sigma. Glutathione Sepharose 4B (17-0756-05) was obtained from GE Healthcare, Monoclonal anti-HA antibody (901503, 1:3000), anti-GAPDH (649201, 1:3000) antibody was obtained from Biolegend.

The polyclonal phosphorylation antibodies against pS549-PDK1 (1:1000) generated by Abclonal Technology were derived from rabbit with four clones. The antigen sequence used for immunization was PDK1 aa544-552 (C-RQRYQ**S**HPD). **S** stands for phosphorylated residue in this synthetic peptide. The antibodies were affinity purified using the antigen peptide column, but they were not counter selected on unmodified antigen.

**shRNAs, sgRNAs and CRISPR/CAS9-mediated knockout assay**. shRNAs against 14-3-3γ were used previously[38]. The lentiviruses for CRISPR/Cas9-based editing of *PDK1* were generated by cloning the annealed short guide RNAs (sgRNAs) into BsmBI-digested lenti-CRISPR V2 vector, which encodes both Cas9 and an sgRNA of interest, as previously described[44]. The sgRNAs were designed by CRISPR Design tool (crispr.mit.edu) as listed below:
  sghPDK1 Forward: 5′-CACCGcaagtttgggaaaatccttg
  Reverse: 5′-AAACcaaggattttcccaaacttgC

**Immunoprecipitation, GST pull-down assays and western blot**. Cells were lysed in EBC buffer (50 mM Tris pH 7.5, 120 mM NaCl, 0.5% NP-40) supplemented with protease inhibitors (Complete Mini, Roche) and phosphatase inhibitors (phosphatase inhibitor cocktail set I and II, Calbiochem). The protein concentrations of whole cell lysates were measured by the Beckman Coulter DU-800 spectrophotometer using the Bio-Rad protein assay reagent. The same amounts of WCL were resolved by SDS-PAGE and immunoblotted with indicated antibodies. For immunoprecipitation analysis, whole cell lysates (WCL) were incubated with the distinct agarose beads for Flag, HA or GST for 2–4 h at 4 ℃. The recovered immuno-complexes products were washed four times with NETN buffer (20 mM Tris, pH8.0, 150 mM NaCl, 1 mM EDTA and 0.5% NP-40) before being resolved by SDS-PAGE and immunoblotted with indicated antibodies.

**Purification of GST-tagged proteins from bacteria**. Recombinant GST-conjugated PDK1 PH domain was generated by transforming the BL21 (DE3) *E. coli* strain with pGEX-PDK1-PH, respectively. The cultured bacteria were grown at 37 ℃ to an O.D. 0.8, and then the protein expression was induced for 12–16 h by adding 0.1 mM IPTG at 16 ℃ with vigorous shaking. Recombinant proteins were purified from harvested pellets and re-suspended in 10 ml EBC buffer for sonication. Insoluble proteins and cell debris were discarded, and the supernatant was incubated with 50 μl 50% Glutathione-sepharose slurry (Pierce) for 3 h at 4 ℃. The Glutathione beads were washed 3 times with PBS buffer and stored at 4 ℃ in PBS buffer containing 10% glycerol or eluted by elution buffer. Recovery and yield of the desired proteins (or complexes) was confirmed by analyzing 10 μl of beads by Coomassie blue staining, and quantified with BSA standards.

**Purification of His-AKT1 protein from insect S9 cells**. AKT1 cDNA was cloned into an insect cell expression vector, pFastBac-HT A (Invitrogen), in frame with his tag at the N-terminus. Constructs were transformed into DH10Bac bacteria (Invitrogen) to generate bacmid DNAs. AKT1 baculoviruses were produced in Sf9 insect cells according to the manufacturer's specification. His-tagged proteins were purified on HisPur Cobalt resin (Thermo Scientific) and eluted by Imidazole (Sigma) as we did previously[44].

**Peptide synthesis and dot immunoblot assays**. PDK1-S549-WT and PDK1-S549-Phospho peptides used for dot blot assays were synthesized by Abclonal Technology. The sequences were listed as below:

PDK1-S549-WT: C-RQRYQSHPD
PDK1-S549-Phospho: C-RQRYQ(pS)HPD

Peptides were diluted into 2 mg/ml for further biochemical assays. For dot blot assays, peptides were diluted with PBS and spotted onto nitrocellulose membrane with the amount of 0.05 μg, 0.15 μg, 0.45 μg and 1.25 μg. The membrane was dried and blocked by soaking in TBST buffer with 5% non-fat milk for immunoblot analysis.

**S6K1, PDK1 and AKT in vitro kinase assays**. 1 μg of the insect cell purified His-AKT1 fusion proteins were incubated with immunoprecipitated PDK1 purified from 293 T cells transfected with various mutant *PDK1* in the presence of ATP in the kinase reaction buffer (50 mM Tris pH 7.5, 1 μM MnCl₂, 2 mM DTT) for 30 min at 30 °C. The reaction was stopped by adding 3 x SDS loading buffer and resolved by SDS-PAGE. The phosphorylation His-AKT1 was detected by specific antibody to recognize pT308-AKT.

**Mass spectrometry analyses**. For mass spectrometry analysis, anti-Flag immunoprecipitations (IP) were performed with the WCL derived from three 10 cm dishes of HEK293 cells transfected with Flag-PDK1. The proteins were resolved by SDS-PAGE, and identified by Coomassie staining. The band containing PDK1 was reduced with 10 mM DTT for 30 min, alkylated with 55 mM iodoacetamide for 45 min, and in-gel-digested with trypsin enzymes. The tryptically digested peptides were desalted with monospin C18 column (SHIMADZU-GL), and then analyzed on an Easy-nLC1200 system equipped with a homemade reverse phase C18 column (75 μm × 300 mm, 1.9 μm). The peptides were separated with a 120 min gradient from 5% to 100% of buffer B (buffer A: 0.1% formic acid in water; buffer B: 0.1% formic acid in 80% Acetonitrile) at 300 nL/min. The eluted peptides were ionized and directly introduced into a Q-Exactive mass spectrometer (Thermo Scientific, San Jose, CA) using a nano-spray source with the application of a distal 2.5-kV spray voltage. A cycle of one full-scan MS spectrum (m/z 300−1500) was acquired followed by top 20 MS/MS events, sequentially generated on the first to the twentieth most intense ions selected from the full MS spectrum at a 30% normalized collision energy.

The acquired MS/MS data were analyzed against a homemade database (including all target proteins) using PEAKS Studio 8.5. Cysteine alkylation by iodoacetamide was specified as fixed modification with mass shift 57.02146 and Methionine oxidation, protein n-terminal acetylation as variable. Additionally, phosphorylation was set as dynamic modification with mass shift 79.9663. In order to accurately estimate peptide probabilities and false discovery rates, we used a decoy database containing the reversed sequences of all the proteins appended to the target database.

**Colony formation assays**. Cells were seeded into 6-well plates (300 or 600 cells/well) and left for 12−20 days until formation of visible colonies. Colonies were washed with PBS twice and fixed with 10% acetic acid/10% methanol for 20 min, then stained with 0.4% crystal violet in 20% ethanol for 20 min. After staining, the plates were washed and air-dried, and colony numbers were counted and quantified.

**Soft agar assays**. The assays were preformed using 6-well plates where the solid medium consists of two layers. The bottom layer contains 0.8% noble agar and the top layer contains 0.4% agar suspended with $1 \times 10^4$ or $3 \times 10^4$ cells. In total 500 μl complete DMEM medium was added every 7 days to keep the top layer moisture and 4 weeks later the cells were stained with iodonitrotetrazolium chloride (1 mg/ml) (Sigma I10406) for colony visualization and counting.

**Immunofluorescence (IF) staining**. A total of 293 T cells transfected with indicated constructs were seeded in chambers. Cells were then fixed with 4% paraformaldehyde for 15 min, followed with 0.1% Triton X-100 in PBS for 15 min. Cells were pre-blocked with 2% BSA for 1 h at room temperature, then incubated with primary antibodies overnight at 4 °C and followed with secondary antibody conjugated with Alexa-fluor-488 or 647. DAPI was used to stain nuclei. Quantification of plasma membrane PDK1 translocation was determined as previously reported[45]. Briefly, the average pixel fluorescence intensity within an area of defined size drawn over 3 distinct areas of the plasma membrane or the cytosol were measured, and displayed as ratios of plasma membrane to cytosolic pixel fluorescence intensity.

For each detection, at least 60 cells were calculated from 3 independent experiments.

**Immunohistochemistry (IHC) staining**. IHC was performed on four micronthick, FFPE sections. FFPE sections were deparaffinized using xylene and rehydrated in graded ethanol. Sections were heated with a pressure cooker to 125 °C for 30 s and 90 °C for 10 s in citrate buffer (pH 6.0) for antigen retrieval. After quenching of endogenous peroxides with 3% H₂O₂ in methanol, all sections were blocked with 3% BSA for 60 min at room temperature. Sections were then incubated with anti-Ki67 (ab16667, 1:200), anti-pS240/244-S6 (CST#5364, 1:500) antibody overnight at 4 °C. Following primary antibody incubation, sections were incubated with monoclonal mouse anti-rabbit immunoglobulins for 60 min at room temperature. Afterwards, sections were developed using the DAB chromogen kit (Dako #K3468) and lightly counterstained with hematoxylin. The score of the IHC signals was judged by two independent pathologists blindly.

**Mouse xenograft assays**. Mouse xenograft assays were performed as described previously[44]. Briefly, $5 \times 10^6$ DLD1-$PDK^{−/−}$ cells stably expressing WT or mutant forms of PDK1 mixed with matrigel were injected into the flank of 6 female nude mice (4-5 weeks of age, were bought from Sun Yat-sen University mouse facility and housed in the specific pathogen-free facilities with 12 h dark/light cycle). Tumor size was measured every three days with a caliper, and the tumor volume was determined with the formula: $L \times W^2 \times 0.5$, where L is the longest diameter and W is the shortest diameter.

**Quantification and statistical analysis**. The in vitro experiments were repeated at least three times unless specifically stated. The compared groups were set similarly in all procedures. For animal studies, we established the number of conditions to test the hypothesis, and two groups were randomly assigned. Results were collected and analyzed blindly. As indicated in the figure legends, all quantitative data are presented as the mean ± SD of three biologically independent experiments or samples; $n$ and $P$-values are indicated in every single figure. Two-way ANOVA was used for multiple group comparisons, and unpaired two-tailed $t$-tests for two-group comparisons. Significant statistical differences between groups were indicated as: *$P < 0.05$; **$P < 0.01$. Statistical analyses and graphics were carried out with GraphPad Prism software 7.0 and Microsoft Excel 16.0.

**Reporting summary**. Further information on research design is available in the Nature Research Reporting Summary linked to this article.

## Data availability

The mass spectrometry-based screening data generated in this study have been deposited in iProX/ProteomeXchange under the accession code IPX0003428000/PXD028167. The genetic alterations of *PDK1* in cBioPortal for Cancer Genomics datasets were integrated from www.cbioportal.org, with the query of gene "PDPK1" for both mutation and copy number alterations (CNA) in different cancer types. Source data are provided with this paper.

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

## Acknowledgements

We thank members of the Guo and Wei laboratory for critical reading and kind suggestions of the manuscript. We thank Dr. Nam Chu in Brigham Women's Hospital at Harvard Medical School for kindly providing the recombinant His-AKT and PDK1 proteins purified from the insect cells. We thank Yilin Li, Ping Wu, and Chao Peng in National Facility for Protein Science in Shanghai for Mass Spectrometry analysis. Figures 4n and 7 were created with BioRender.com. This work was supported by China National Natural Science Foundation (J.G. 31871410, 32070767; Q.J. 32100559), China Postdoctoral Science Foundation (Q.J. 2020M683035).

## Author contributions

Conception and design: J.G., Q.J., W.W.; Development of methodology: Q.J., X.Z.; Acquisition of data (provided animals, acquired and managed patients, provided facilities, etc.): Q.J., X.Z., N.Z., S.H., X.W., L.W.; Analysis and interpretation of data (e.g., statistical analysis, biostatistics, computational analysis): Q.J., J.G., N.Z.; Writing the manuscript: Q.J., J.G.; Administrative, technical, or material support (i.e., reporting or organizing data, constructing databases): Q.J., X.D., X.Z., S.H., L.W., N.Z., W.X.; Study supervision: W.X., J.G. Approved manuscript: all authors.

## Competing interests

W.W. is a co-founder and consultant for the ReKindle Therapeutics. Other authors declare no competing interests.
