## [Peer Review File · Nature Communications]

S6K1-mediated phosphorylation of PDK1 impairs AKT kinase activity and oncogenic functionsREVIEWER COMMENTS

Reviewer #1 (Remarks to the Author):

In this manuscript, the authors identify a novel feedback mechanism of PDK1 inactivation that suppresses AKT signalling and tumor growth. The authors demonstrate that S6K1 phosphorylates PDK1 at S549, which inhibits PDK1 binding to PI(3,4,5)P3 and prevents AKT phosphorylation and activation. Mechanistically, PDK1 phosphorylation enhanced its interaction with 14-3-3 to promote PDK1 dimerization and sequester it from AKT. PDK1 mutations near the S549 phosphorylation site prevented PDK1 phosphorylation leading to enhanced AKT T308 phosphorylation, cellular transformation and tumor growth. This study reveals a regulatory mechanism for PDK1-mediated AKT activation, which will be of interest to the cancer cell signaling field. However, there is insufficient information regarding the novel PDK1 mutations identified in the manuscript, which limits the clinical impact of the study.

Major comments

1. There is no description of how frequently the human PDK1 mutations occur and which cancers these mutations are observed in. It is unclear whether the DLD1 colon cancer cells are a clinically relevant model to examine the tumorigenic potential of these mutants. The R544K, R546P, S549N and P551L mutations do not appear to be from TCGA cohorts as the authors propose in their manuscript. Could this be clarified and the frequency of these mutations and the cancers in which they occur provided in more detail.
2. This study relies heavily on only one colon cancer cell line (PDK1-KO DLD1 cells) to demonstrate the oncogenic function of PDK1 mutations. The authors need to validate the effects of PDK1 mutations on colony formation in another independent cancer cell line, ideally one that is representative of the clinical subset of patients that exhibit PDK1 mutations.
3. As PDK1 can phosphorylate a number of other targets such as S6K, SGK and RSK (PMID: 28473254), the authors need to demonstrate that mutant PDK1 exerts its oncogenic function via AKT by examining whether the increased colony formation can be rescued by AKT inhibition or knock-down. In addition throughout the manuscript the effects on Thre-phosphorylation of AKT are shown but not quantitated, this should be undertaken.
4. Some of the data examining PDK1 localization to the plasma membrane is not very convincing. For example, the authors claim that LY294002 could dramatically block WT and S549A PDK1 plasma membrane recruitment (Supplementary Fig. 1C), however there appears to be clear plasma membrane staining in the LY204002-treated samples. Moreover, in Fig. 3I WT and S549A PDK1 appear to show strong cytoplasmic staining despite the authors proposing in Supplementary Fig. 1C that they are plasma membrane associated. Further quantification of this data is needed to substantiate the authors conclusions. ie for all images the plasma membrane to cytosol ratio needs to be appropriately quantitated.
5. The authors suggest that "both type I and II mutants attenuated PDK1-14-3-3 γ interaction (Fig. 5C), resulting in increased PDK1 binding to PIP3 (Fig. 5D), and their membrane localization (Fig. 5E and 5H). However, Fig. 5D only shows that type II mutants bind to PIP3, there are no type I mutants in this experiment. Further, in Fig. 5E there is little to no difference in the amount of type I/II versus WT PDK1 in the membrane fractions, this would need to be quantified to demonstrate any meaningful difference here.

Other comments

1. The top of Supplementary Fig 1B is cut off so it is impossible to identify the corresponding samples and verify the findings.

2. In Supplementary Fig. 1c, the authors use LY294002 treatment to investigate PI3K-recruitment of PDK1 (Supplementary 1C), however, LY294002 also has off-target effects on other PI3K and non-PI3K proteins (PMID 17302559). The highly specific PI3K inhibitor BKM120 would be more appropriate for this assay.
3. In Fig. 1B, the second blot from the top does not appear to match up with the rest of the blots, there is a faint band that spans across lanes 2 and 3.
4. In Fig. 1C, the correct control for the inhibitor treatment depicted should be a vehicle control and not empty vector as indicated on the figure.
5. The authors claim in Fig. 1F that “insulin could not accelerate PDK1 phosphorylation in S6k1 and S6k2 double knockout (S6k1/2^{-/-}) MEFs, which could be robustly induced by re-introduction of S6K1 (Fig. 1F)”. However, insulin is clearly having an effect on PDK1 phosphorylation in S6K1/2 knockout cells as there are bands in the insulin-treated samples that are not present with insulin absence.
6. On the left hand side of Fig. 1G the label should be IP:PDK1 and not IB:PDK1. There appears to be less immunoprecipitated PDK1 in S6K1 shRNA samples. The authors should blot PDK1 in the whole cell lysate to make sure PDK1 protein levels aren't affected by S6K1 shRNA.
7. In Fig. 2B and Supplementary Fig. 4A, these samples appear to have been run across multiple immunoblots. Can the authors clarify how these samples were processed to ensure that the samples from different gels are directly comparable?
8. Fig. 2I, 6I and 6J depict relative Ki67 and pS6 staining. Can the authors clarify what this is relative too? A better way to express this is as percentage of Ki67 or pS6 positive cells.
9. In Supplementary Fig. 7, the labels on the right hand side of figures are cut off.
10. The authors suggest in Supplementary Fig. 7I that “depletion of endogenous 14-3-3 γ could significantly enhance PDK1-PIP3 interaction”. However, PDK1 association with PIP3 looks unchanged with 14-3-3 γ shRNA in this figure.

Reviewer #2 (Remarks to the Author):

Jiang et al. have identified a S6K1-mediated phosphorylation site in the vicinity of the PH domain of PDK1. The phosphorylation of Ser549 in PDK1 attenuates PIP3 binding and AKT phosphorylation on Thr308 by disrupting the PIP3-induced proximity of the two PI3K-downstream kinases. Furthermore, the authors show a pSer549-dependent interaction of PDK1 and 14-3-3, a complex which further disrupts AKT engagement. The authors then also claim that PDK1-(14-3-3) homodimerizes and still phosphorylates and activates S6K1 to enforce a negative feedback loop. The findings are important and topical and could contribute to a refined understanding of the PI3K/mTOR pathway.

Major points:

General:

- 1) While most statements are qualitatively supported by data, too much data is only presented as n=1 (for example western blots, imaging data). The whole work has to be revised to quantify western blot and imaging data. For western blot data, the pertinent bands have to be quantified and expressed as phosphorylated protein / non-phosphorylated protein (or if not possible as ratio over loading control protein). Image data has to be quantified by non-biased, user-independent algorithms, which have to be

described in methods.

The quantitated data has then to be statistically analyzed (see 2).

2) Statistics: an undefined "t-test" (Paired? Unpaired? Single sided?) and an ANOVA test (where was this applied?) is mentioned in the methods part. Data should be analyzed by non-parametric statistical tests, not "t-test". All the data have to be re-analyzed.

3) Figure labelling is poorly designed. Lack of explanation of abbreviations (EV, WLC, etc.) makes evaluation of figures guesswork. Labelling is often contradictory and wrong/sloppy: see 2G: IB: PDK1 left and contradictory IB: x on the right. Reading occurs from left to right, put important labels on the left of images. Remove repetitive labels by better figure design (IB, IB, IB...).

Specific:

4) An important point to understand the role of PDK1 phosphorylation on S549 is the quantification of the fraction of PDK1 that is phosphorylated during a surface receptor ligand (insulin, growth factor...) stimulation. If this fraction is low (5% of PDK1 phosphorylated), this phosphorylation would very likely constitute a (rapid ?) deactivation process of engaged PDK1. If the fraction of phosphorylated PDK1 is >90% this would point to a complete arrest of the relay of PI3K signaling to TORC1. The authors should thus provide an estimation (by MS or standardized IBs) how much of the total PDK1 is actually phosphorylated. In this context, the authors should also provide a time curve of the PDK1 S549 phosphorylation. As most of the PDK1 S549 phosphorylation is induced by overactivated kinase constructs (myr-Akt, S6K1-R3A, etc.) throughout the work, the use of other growth factors apart from insulin would also add confidence that this observation can be generalized.

5) The general conclusion suggests that an activated PI3K/mTOR pathway would lead to an increase of PDK1-pS549. The investigation of some cell lines with constitutively activated PI3Kalpha (e.g. 1047 mutation [SKOV3, T47D, etc. cell lines] or loss of PTEN (PC3, A2058, etc.) would show if PDK1-pS549 constitutively associates with increased PIP3, or if adaptation occurs.

6) Where in the cell does S6K1 phosphorylate PDK1? TORC1 is active on endo-membranes (late endosomes) and the PIP3-activated PDK1 is assumed to operate at the plasma membrane.

7) Figure 1: In C at least one pan-PI3K inhibitor should be used. As mentioned above for all items in Fig. 1 at least pS549, RxRxxpS, pT308-Akt and pS6 should be quantified (same for all figures) and statistically analysed.

J. nor readable. Move to supplementary and present full page (could not be evaluated).

8) Figure 2: quantifications and statistics as requested above. I: no sufficient image resolution, too small. Show enlarged example cut-outs.

9) Figure 3: Add quantifications and statistics (WB and images).

10) Figure 4: GST-PD should read GSH(beads)-PD. And explain abbreviations.

a) the claimed selectivity for the 14-3-3 gamma isoform binding to PDK1 is somewhat astonishing. Is there a structural determinant that would be responsible for that? What is the status of the other 14-3-3 isoforms? Band shifts seem to propose post-translational modification – and inactivation of binding?

b) L: as mentioned above an insulin pPDK1 time curve should be included.

c) There is no convincing data presented to warrant the proposal of a pPDK1-(14-3-3)-pPDK1 complex. Here size exclusion FPLC or similar would be required.

11) Figure 5: Quantification, labelling. This is interesting information.

12) Figure 6: B: at the limit of resolution. Please adapt statistics. For F provide point by point statistical analysis (non-parametric) and 2-way ANOVA for the whole curve.

13) Figure 7: please integrate the established feedback loops emerging from S6K1.

Minor stuff, terminology

- L300: mTROC2: typo

- Use "cellular" where experiments were done in cells, use in vivo for results from tumor models

- Define ALL abbreviations

Reviewer #3 (Remarks to the Author):

While it has long been appreciated the major role of AKT activation in cancer, upstream regulation of PDK1, which phosphorylates AKT at T308 is not fully understood. The authors identify a novel inhibitory phosphorylation site on PDK1 (S549) by S6K1 and propose a mechanism by which phosphorylation of this site decreases PDK1 membrane localization and allows 14-3-3y binding to promote its dimerization and prevent further activity towards AKT T308. They show mutations in and around this site promote tumorigenicity in cells and animals.

These studies are quite thorough and do a good job of demonstrating the PDK1 S549 is important for modulation of PDK1 activity. However, there are a few additional or updated figure panels will support your hypothesis more fully.

Major comments:

As the p-PDK1 S549 antibody still binds with PDK1 S549A, please show validation of this antibody using a phosphatase with WT PDK1 to show decreased signal.

Please add in an experiment where P551A mutant and 14-3-3y KD are done at the same time and cells are stimulated with insulin to show kinetics of AKT T408 phosphorylation under these conditions.

Nearly every Figure Legend is missing some very relevant information (N, how data was normalized). Interestingly, the authors did n = 5 mouse tumors for the growth assays but did not stain all 5 tumors for IHC or run all 5 on WB. I would like to see the data for all 5.

Figure 7 Legend – Please add more information about the meaning of red v blue arrows, solid v dotted lines, and a more comprehensive explanation of the pathways shown.

Minor comments:

Please check grammar and tense of verbs used throughout the publication, especially usage of the word "could".

Figure S1B – WT-PDK1 appears to be in the nucleus in this panel but not in S1C/E. Is this a representative image or can you choose another cell to better show the localization?

Fig. 1- E) Show HA blot. F) S6K1 is missing its "1". G) Please show an IgG control. "IB:PDK1" should read "IP:PDK1".

Line 128 – Insulin did enhance PDK1 phosphorylation in S6k1/2-/- MEFs. Please reflect this in the text.

Line 131-132 – "depletion of S6K1 in cancer cells could also reduce PDK1 phosphorylation" -please cite this data

Line 135 – The S459A mutant form had reduced phosphorylation (Fig. 1I), not completely abrogated as the text states.

Fig. S2A,B – Please show a lower exposure IP: Flag; IB:Flag blot (A) and WCL IB:Flag blot (B).

Fig. 2A – The text says that AKT p-T308 is significantly elevated. Please show quantitation of this.

Fig. 2B – Can you run the WT and SA mutant on the same blot so we can accurately compare their signaling?

Fig. 2G/I – Are any other points significantly different?

Line 168 – Add Fig. 2E to the parentheses here

Fig. 2 - Can you explain the discrepancy on p-AKT S473 seen in Fig. 2B versus Fig. 2H? Effect of in vivo tumor growth versus cell culture?

Can you discuss why WT PDK1 does not seem to show any significant enhancement in tumorigenicity or cell growth phenotypes over empty vector? (Ex. Fig. S4E,F and Fig. 6)

Fig. S5 – Please use "N.S." instead of "#" for $P > 0.05$

Fig. S6H – Can you add another couple lanes to this experiment where you add one WT and one SA/SD? Can you also show 14-3-3y in this figure?

Fig 3G – Please add S6K1 total

Fig. S7C/; Fig. 4G – Please add p-PDK1 blot to show if the P551A mutant can still be phosphorylated.

Fig. S7F – Please use lower exposure blot for GST.

Fig. 4E – Please show lower exposure of both Flag blots.

Fig. 4I – What cell type is this? This one AKT activation blot looks different than the others - why doesn't WT activate AKT T308 until 30 minutes?

Fig. 4J – Please add 14-3-3y to the PD blot.

Line 254-255 – Fig. S7I doesn't support your statement that 14-3-3y significantly enhances the interaction. Please quantify this over several experiments.

Fig. 5 – Please continue the Type I and Type II labels throughout the figure. Please add Type I mutants to panel D. Add another experiment where you pulldown Flag and probe for 14-3-3y (opposite pulldown to 5C).

Fig. 5F – please quantify the "enhanced interaction" between AKT and PDK1 or mention how reproducible this is.

Fig. 5G – Show lower exposure for pT308-AKT and add TI and TII labels. Both S549N and P551Q are not really convincing in their "enhanced AKT activation". You may want to show other substrates of AKT to make this more convincing such as PRAS40.

Lines 267-269 – Please give more information about how frequent these mutations are in cancer, what types of cancer they occur in, do tumors ever have type I and type II together, and any other relevant information about this analysis.

Line 326-328 – Please elaborate on why you think 14-3-3t and pS241-PDK1 might have an effect on the 14-3-3/pS549 interaction you demonstrate in the this paper or delete this postulation.

Point by Point Rebuttal

(Manuscript number: NCOMMS-21-18003)

We sincerely appreciate the thorough analyses and constructive comments provided by the three reviewers, which have been very helpful in guiding us to further improve our manuscript. As described in details below, we have fully addressed the concerns raised by the reviewers.

Reviewer #1 (Remarks to the Author):

In this manuscript, the authors identify a novel feedback mechanism of PDK1 inactivation that suppresses AKT signalling and tumor growth. The authors demonstrate that S6K1 phosphorylates PDK1 at S549, which inhibits PDK1 binding to PI(3,4,5)P3 and prevents AKT phosphorylation and activation. Mechanistically, PDK1 phosphorylation enhanced its interaction with 14-3-3 to promote PDK1 dimerization and sequester it from AKT. PDK1 mutations near the S549 phosphorylation site prevented PDK1 phosphorylation leading to enhanced AKT T308 phosphorylation, cellular transformation and tumor growth. This study reveals a regulatory mechanism for PDK1-mediated AKT activation, which will be of interest to the cancer cell signaling field. However, there is insufficient information regarding the novel PDK1 mutations identified in the manuscript, which limits the clinical impact of the study.

Response: We thank the reviewer in recognizing the novelty and the potential impacts of our study. We also appreciate the constructive comments from the reviewer to help us to further strengthen our manuscript. Below we also provided more information regarding the novel *PDK1* mutations identified in the revised manuscript.

Major comments

1. There is no description of how frequently the human PDK1 mutations occur and which cancers these mutations are observed in. It is unclear whether the DLD1 colon cancer cells are a clinically relevant model to examine the tumorigenic potential of these mutants. The R544K, R546P, S549N and P551L mutations do not appear to be from TCGA cohorts as the authors propose in their manuscript. Could this be clarified and the frequency of these mutations and the cancers in which they occur provided in more detail.

Response: We thanks the reviewer for raising this great question. We fully agree with the reviewer that it is very important for future readers to access to the description of how frequent the human *PDK1* mutations occur and which cancers these mutations are observed, both of which are every important information. To this end, we summarized these mutations and observed that mutations mainly occurred in colon adenocarcinoma (**Fig. R1**). Thus, we chose a colon cancer cell DLD1 with an intact *PDK1*, among which *PDK1* has been genetically depleted as previously reported as a cell-culture model (Kajsa Ericson, et al, *PNAS*, 2010). In this cell line, we re-introduced different *PDK1* patient-derived mutations including R544K, R546P, S549N and P551L, to investigate their oncogenic functions compared with WT *PDK1* (**revised Fig. 6**). To further validate our findings, we also chose another colon cancer cell line SW480 to study the potential oncogenic function of these *PDK1* mutations and obtained a similar result (**revised Fig. S8d-e**). These newly obtained data together suggest that *PDK1* mutations at least in colon cancer could promote cell malignancies partially by activating the AKT kinase.

TCGA-EM-A2CJ...	Papillary Thyroid Cancer	P551Q	○	Missense
TCGA-FS-A1Z4-06	Cutaneous Melanoma	P551Q	○	Missense
PECAPJ41CLON...	Esophageal Poorly Differentiated ...	P551L	○	Missense
PECAPJ41CLON...	Esophageal Poorly Differentiated ...	P551L	○	Missense
TCGA-G2-A2EO...	Bladder Urothelial Carcinoma	P551A	○	Missense
coadread_dfci_2...	Colorectal Adenocarcinoma	S549N	○	Missense
P-0006793-T01-I...	Lung Adenocarcinoma	R546P	○	Missense
P-0004795-T01-I...	Colon Adenocarcinoma	R544K	○	Missense

Fig. R1. The mutant information for *PDK1* in different cancer types and its frequency by analyzing the TCGA database (<http://www.cbioportal.org/public-portal/>). As indicated, two Type I mutations including R544K, S549N were observed in colorectal adenocarcinoma. Thus, we chose colon cancer cell lines including DLD1 and SW480 with intact *PDK1* to further investigate mutant *PDK1*'s oncogenic functions.

2. This study relies heavily on only one colon cancer cell line (PDK1-KO DLD1 cells) to demonstrate the oncogenic function of PDK1 mutations. The authors need to validate the effects of PDK1 mutations on colony formation in another independent cancer cell line, ideally one that is representative of the clinical subset of patients that exhibit PDK1 mutations.

Response: Since DLD1 colon cancer cells with intact *PDK1*, genetically knockout *AKT* and *PDK1* were widely used for studies on the oncogenic role of the PI3K-AKT pathway (Kajsa Ericson, et al, *PNAS*, 2010; Jianping Guo, et al, *Science*, 2016). As the reviewer kindly suggested, we tried to employ another cell line that is representative of the clinical subset of patients that exhibit *PDK1* mutations, however, due to the relatively lower occurred frequency, we could not obtain this kind of cell lines. Following the reviewer's kind instruction, we re-introduced *PDK1* mutations into another colon cancer cell line SW480, in which the intact *PDK1* was depleted and different types of mutant *PDK1* were stably expressed. Using these newly generated important cell lines, we

found that, similar to DLD1 cells, these mutations could significantly elevate AKT-pT308, as well as colony formation capabilities in WE480 cells (**revised Fig. S8d-e**).

3. As PDK1 can phosphorylate a number of other targets such as S6K, SGK and RSK (PMID: 28473254), the authors need to demonstrate that mutant PDK1 exerts its oncogenic function via AKT by examining whether the increased colony formation can be rescued by AKT inhibition or knock-down. In addition, throughout the manuscript the effects on The-phosphorylation of AKT are shown but not quantitated, this should be undertaken.

Response: As the reviewer kindly mentioned, PDK1 can phosphorylate a number of other AGC targets such as S6K, SGK and RSK. Thus we have measured PDK1 functions (WT, S549A, S549D) under the conditions of depletion *AKT1* or treated with the AKT1 inhibitor. We observed that PDK1 function is mainly via modulating AKT1 kinase (**revised Fig. S5a-c**). As kindly suggested, we also performed similar experiments to validate whether mutant *PDK1* exerts its oncogenic function via activating the AKT kinase. To this end, we knocked down *AKT1* in type I or type II *PDK1* mutant-expressing DLD1-*PDK1*^{-/-} cells (**revised Fig. S9a**), and observed that mutant *PDK1* expression induced colony formation could be abrogated by knockdown *AKT1* (**revised Fig. S9b-c**). These findings together indicate that patients-derived mutant *PDK1* exerts its oncogenic roles mainly via activating the AKT oncogenic kinase.

As kindly suggested, we have quantified the phosphorylation of Akt with Image J in major experiments (**revised Fig. 1c, 2a, 6a, S2a, S8d**).

4. Some of the data examining PDK1 localization to the plasma membrane is not very convincing. For example, the authors claim that LY294002 could dramatically block WT and S549A PDK1 plasma membrane recruitment (Supplementary Fig. 1C), however there appears to be clear plasma membrane staining in the LY204002-treated samples. Moreover, in Fig. 3I WT and S549A PDK1 appear to show strong cytoplasmic staining despite the authors proposing in Supplementary Fig. 1C that they are plasma membrane associated. Further quantification of this data is needed to substantiate the authors conclusions. ie for all images the plasma membrane to cytosol ratio needs to be appropriately quantitated.

Response: As kindly suggested, we have repeated these staining, and obtained consistent findings which showed that PDK1 was mainly localized in the cytoplasm under normal condition or treated with the PI3K specific inhibitor BKM120 (**revised Fig. S1e**). On the other hand, PDK1 stimulated with insulin or harbored non-phosphorylation mutant S549A or other patients derived mutations including R544K ,R546P, S549N, P551A, P551L, P551Q displayed markedly elevation of membrane locations, which has been further quantified for the plasma membrane to cytosol ratio (**revised Fig. 3h, 5f, S1d-e, S1g, S6f**).

5. The authors suggest that “both type I and II mutants attenuated PDK1-14-3-3 γ interaction (Fig. 5C), resulting in increased PDK1 binding to PIP3 (Fig. 5D), and their membrane localization (Fig. 5E and 5H). However, Fig. 5D only shows that type II mutants bind to PIP3, there are no type I mutants in this experiment. Further, in Fig. 5E there is little to no difference in the amount of type I/II versus WT PDK1 in the membrane fractions, this would need to be quantified to demonstrate any meaningful difference here.

Response: As kindly suggested, we have included data with type I mutants in the **revised Fig. 5d**. Furthermore, we have obtained the clean data to show the difference of type I/II mutant *PDK1* versus WT PDK1 in their location in membrane (**revised Fig. 5e**), and quantified the PDK1 blot compared with the Tubulin (for cytoplasm protein) or AIF (for membrane protein) (**revised Fig. 5e**).

Other comments

1. The top of Supplementary Fig 1B is cut off so it is impossible to identify the corresponding samples and verify the findings.

Response: We have obtained the suitable **revised Fig S1d** without cut off its top.

2. In Supplementary Fig. 1c, the authors use LY294002 treatment to investigate PI3K-recruitment of PDK1 (Supplementary 1C), however, LY294002 also has off-target effects on other PI3K and non-PI3K proteins (PMID 17302559). The highly specific PI3K inhibitor BKM120 would be more appropriate for this assay.

Response: As kindly instructed, we have employed the specific PI3K inhibitor BKM120 to repeat the experiments performed with LY294002, and found that BKM120 also dramatically blocked PDK1-WT and S549A membrane translocation (**revised Fig. S1e**).

3. In Fig. 1B, the second blot from the top does not appear to match up with the rest of the blots, there is a faint band that spans across lanes 2 and 3.

Response: As kindly instructed, we have aligned the top blot and matched up with the rest of the blots (**revised Fig. 1b**).

4. In Fig. 1C, the correct control for the inhibitor treatment depicted should be a vehicle control and not empty vector as indicated on the figure.

Response: We thank the reviewer for pointing out this mislabeling. We have corrected this EV to DMSO, which acted as a vehicle control for these inhibitors (**revised Fig. 1c**).

5. The authors claim in Fig. 1F that “insulin could not accelerate PDK1 phosphorylation in S6k1 and S6k2 double knockout (S6k1/2^{-/-}) MEFs, which could be robustly induced by re-introduction of S6K1 (Fig. 1F)”. However, insulin is clearly having an effect on PDK1 phosphorylation in S6K1/2 knockout cells as there are bands in the insulin-treated samples that are not present with insulin absence.

Response: We thank the reviewer for raising this important point. Actually, insulin indeed induced PDK1 phosphorylation in *S6k1/2^{-/-}* MEFs detected with the anti-RxRxxpS antibody, indicating that there would likely be other AGC kinase involved in PDK1 phosphorylation or the background phosphorylation. I hope the reviewer would concur with us that S6K1 was the major kinase to promote PDK1 phosphorylation at S549 (**revised Fig. 1f, 1i**).

6. On the left hand side of Fig. 1G the label should be IP:PDK1 and not IB:PDK1. There appears to be less immunoprecipitated PDK1 in S6K1 shRNA samples. The authors should blot PDK1 in the whole cell lysate to make sure PDK1 protein levels aren't affected by S6K1 shRNA.

Response: We are sorry for a wrong label that it should be IP: PDK1 instead of IB: PDK1 on the left hand side of Fig. 1G. Following the kind suggestion from the reviewer, in this experiment, we also included an IgG IP as a negative control, and also include the PDK1 blot in the whole cell lysate to show the expression of PDK1. Meanwhile, the relative of PDK1 phosphorylation (RxxpS and pS549) was quantified as suggested (**revised Fig. 1g**).

7. In Fig. 2B and Supplementary Fig. 4A, these samples appear to have been run across multiple immunoblots. Can the authors clarify how these samples were processed to ensure that the samples from different gels are directly comparable?

Response: It is important to compare these findings in the same gel as the reviewer kindly mentioned. Actually, we have transferred the gel into the same membrane, which has been further

subjected for WB to keep the same conditions and exposure (**revised Fig. 2b, S4a**). By the way, we also run these proteins in the same gel and obtained a similar result (**Fig. S4b**).

8. Fig. 2I, 6I and 6J depict relative Ki67 and pS6 staining. Can the authors clarify what this is relative too? A better way to express this is as percentage of Ki67 or pS6 positive cells.

Response: As kindly suggested, we have enlarged and re-quantified the original Fig2I, 6I, and 6J data as percentage of Ki67 or pS6 positive cells (**revised Fig. 2i, S4g, S8j-m**).

9. In Supplementary Fig. 7, the labels on the right hand side of figures are cut off.

Response: We have corrected this error.

10. The authors suggest in Supplementary Fig. 7I that “depletion of endogenous 14-3-3 γ could significantly enhance PDK1-PIP3 interaction”. However, PDK1 association with PIP3 looks unchanged with 14-3-3 γ shRNA in this figure.

Response: As kindly instructed, we chose a shorter exposure band, which showed a markedly reduction of PDK1/PIP₃ binding upon 14-3-3 depletion, and we further quantified the PIP₃ binding PDK1 compared with input of PDK1 (**revised Fig. S7i**).

Reviewer #2 (Remarks to the Author):

Jiang et al. have identified a S6K1-mediated phosphorylation site in the vicinity of the PH domain of PDK1. The phosphorylation of Ser549 in PDK1 attenuates PIP3 binding and AKT phosphorylation on Thr308 by disrupting the PIP3-induced proximity of the two PI3K-downstream kinases. Furthermore, the authors show a pSer549-dependent interaction of PDK1 and 14-3-3, a complex which further disrupts AKT engagement. The authors then also claim that PDK1-(14-3-3) homodimerizes and still phosphorylates and activates S6K1 to enforce a negative feedback loop. The findings are important and topical and could contribute to a refined understanding of the PI3K/mTOR pathway.

Response: We thank the reviewer in recognizing the novelty and the potential impacts of our study. We also appreciate the constructive comments from the reviewer to help us to further strengthen our manuscript.

Major points:

General:

1) While most statements are qualitatively supported by data, too much data is only presented as n=1 (for example western blots, imaging data). The whole work has to be revised to quantify western blot and imaging data. For western blot data, the pertinent bands have to be quantified and expressed as phosphorylated protein / non-phosphorylated protein (or if not possible as ratio over loading control protein). Image data has to be quantified by non-biased, user-independent algorithms, which have to be described in methods. The quantitated data has then to be statistically analyzed (see 2).

Response: As kindly suggested, we have provided a statistical analysis section and further provided statistical details in the figure legends and method part. Meanwhile, we also quantified the key western blots with phosphorylated protein/non-phosphorylated protein or loading control protein (revised Fig. 1c-h, 2a, 3g, 5e, 6a, S2a-b, S3f, S7i), and imaging data (revised Fig. 2i, 2h, 5f, S1d-e, S1g, S4g, S6f, S8k, S8m).

2) Statistics: an undefined “t-test” (Paired? Unpaired? Single sided?) and an ANOVA test (where was this applied?) is mentioned in the methods part. Data should be analyzed by non-parametric statistical tests, not “t-test”. All the data have to be re-analyzed.

Response: As kindly suggested, we have re-analyzed all the data by using non-parametric statistical tests, which also has been described in the revised method section (**revised Fig. 2e, 2i, 6f, 6i-j, S4g, S8k-m, S9c**).

3) Figure labelling is poorly designed. Lack of explanation of abbreviations (EV, WLC, etc.) makes evaluation of figures guesswork. Labelling is often contradictory and wrong/sloppy: see 2G: IB: PDK1 left and contradictory IB: x on the right. Reading occurs from left to right, put important labels on the left of images. Remove repetitive labels by better figure design (IB, IB, IB...).

Response: As kindly suggested, we have defined all abbreviations in the revised manuscript, and put important labels on the left of images to make it better reading.

Specific:

4) An important point to understand the role of PDK1 phosphorylation on S549 is the quantification of the fraction of PDK1 that is phosphorylated during a surface receptor ligand (insulin, growth factor...) stimulation. If this fraction is low (5% of PDK1 phosphorylated), this phosphorylation would very likely constitute a (rapid ?) deactivation process of engaged PDK1. If the fraction of phosphorylated PDK1 is >90% this would point to a complete arrest of the relay of PI3K signaling to TORC1. The authors should thus provide an estimation (by MS or standardized IBs) how much of the total PDK1 is actually phosphorylated.

Response: We totally agree with the reviewer's excellent point that the different fraction of phosphorylated PDK1 will result in different response. As kindly suggested, we have stimulated cells with insulin, and harvested the cell lysate for MS analysis for PDK1 S549 phosphorylation. We observed that nearly 1/6 of PDK1 was phosphorylated at S549 among the trapped peptides (**revised Fig. S1a**). Furthermore, we also stimulated the cells with insulin or EGF to investigate the response of pS549-PDK1, and observed the fluctuation of pS549-PDK1 (**revised Fig. S3e**). Thus, these observations indicate that this reaction should be a rapid response and likely be regulated for the AKT signaling.

In this context, the authors should also provide a time curve of the PDK1 S549 phosphorylation. As most of the PDK1 S549 phosphorylation is induced by overactivated kinase constructs (myr-Akt, S6K1-R3A, etc.) throughout the work, the use of other growth factors apart from insulin would also add confidence that this observation can be generalized.

Response: As kindly suggested, we stimulated cells with insulin and EGF under physiological conditions, respectively (**revised Fig. S2c and S3e**). Notably, we found that the phosphorylation level of PDK1-S549 increased first with these stimulations and then decreased at the late time of stimulation (**revised Fig. S3e**).

5) The general conclusion suggests that an activated PI3K/mTOR pathway would lead to an increase of PDK1-pS549. The investigation of some cell lines with constitutively activated PI3K α (e.g. 1047 mutation [SKOV3, T47D, etc. cell lines] or loss of PTEN (PC3, A2058, etc.) would show if PDK1-pS549 constitutively associates with increased PIP3, or if adaptation occurs.

Response: As kindly suggested, we have employed the PI3K inhibitor BKM120, which could dramatically block PDK1-pS549 in T47D (PIK3A-H1047R mutation) and PC3 cells (*PTEN* deletion) (**revised Fig. R1**).

Fig. R1 PI3K specific inhibitor BKM120 could dramatically decrease pS549-PDK1 in *PIK3CA*-H1047R mutant T47D and *PTEN* deleted PC3 cell.

6) Where in the cell does S6K1 phosphorylate PDK1? TORC1 is active on endo-membranes (late endosomes) and the PIP3-activated PDK1 is assumed to operate at the plasma membrane.

Response: As we have showed that only around 1/6 PDK1 undergoing pS549 phosphorylation (**revised Fig. S1a**), pS549-PDK1 would likely be a dynamic process (**revised Fig. S3e**). Although mTORC1 is active on endo-membranes to further activate S6K1, in turn the activated S6K1 would phosphorylate PDK1 mainly in the cytoplasm, which would subsequently block PDK1 PIP₃ binding and membrane location. Alternatively, the activated S6K1 also possibly phosphorylates membrane located PDK1, to dissociate PDK1 from PIP₃ binding and membrane location, serving to impair PDK1-mediated AKT-pT308 and its oncogenic function. These important direction is worth to be further investigated. However, we hope the reviewer agree with us that this lies outside the major scope of this manuscript and warrant further studies in a separate manuscript in the future.

7) Figure 1: In C at least one pan-PI3K inhibitor should be used. As mentioned above for all items in Fig. 1 at least pS549, RxRxxpS, pT308-Akt and pS6 should be quantified (same for all figures) and statistically analysed. J. nor readable. Move to supplementary and present full page (could not be evaluated).

Response: As kindly suggested, we employed the PI3K inhibitors LY290004 and BKM120 in this panel of experiments (**revised Fig. 1c**), and quantified phosphorylation of pS549, RxRxxpS, pT308-Akt and pS6 upon the according protein (**revised Fig. 1c**). As kindly instructed we have removed the MS data of original Fig. 1J in the revised figure.

8) Figure 2: quantifications and statistics as requested above. I: no sufficient image resolution, too small. Show enlarged example cut-outs.

Response: As kindly suggested, we have quantified the key blots (**revised Fig. 2a-b, 2h-i**). We also showed the enlarged example cut-out for IHC staining for pS6 (**revised Fig. 2i**) and Ki67 (**revised Fig. S4g**). Meanwhile, the percentage of pS6 or Ki67 positive staining cells was quantified (**revised Fig. 2i and S4g**).

9) Figure 3: Add quantifications and statistics (WB and images).

Response: As kindly suggested, we have generated the quantifications and statistics for the key western bolts and IF images (**revised Fig. 3a-h**).

10) Figure 4: GST-PD should read GSH(beads)-PD. And explain abbreviations.

Response: As kindly suggested, we have explained GSH(beads)-PD as GST-PD, and included all the abbreviation in the revised manuscript.

10a) the claimed selectivity for the 14-3-3 gamma isoform binding to PDK1 is somewhat astonishing. Is there a structural determinant that would be responsible for that? What is the status of the other 14-3-3 isoforms? Band shifts seem to propose post-translational modification – and inactivation of binding?

Response: We thank the reviewer for raising this excellent question. Due to the technic restriction, we could not obtain the structure information about the binding of 14-3-3 γ with the PDK1-PH, and we will devote more efforts for this good suggestion in future, in a separate following up manuscript. Since 14-3-3 γ was the major binding isoform of 14-3-3 in our screen binding system (**revised Fig. 4a**), 14-3-3 τ also has been reported to binding pS241-PDK1, which will also attenuate PDK1 functions (S. Sato, et al, *JBC*, 2002). Whether and how other members of 14-3-3 protein involve in PDK1-AKT signaling regulation are worth to be further studied. As the reviewer mentioned, we indeed observed the band shift in the endogenous binding of PDK1 with 14-3-3 γ (**revised Fig. 4c**), possibly suggests a post-translational modification and inactivation of binding, which need to be further investigated. However, we hope the reviewer agree with us that this lies outside the major scope of this manuscript and warrant further studies in a separate manuscript in the future.

10b) L: as mentioned above an insulin pPDK1 time curve should be included.

Response: As kindly suggested, we stimulated the cells with insulin and EGF under physiological conditions, respectively (**revised Fig. S2c and S3e**). Notably, we found that the phosphorylation level of PDK1-S549 increased first with these stimulations and then decreased at the late time of stimulation (**revised Fig. S3e**).

10c) There is no convincing data presented to warrant the proposal of a pPDK1-(14-3-3)-pPDK1 complex. Here size exclusion FPLC or similar would be required.

Response: As kindly suggested, we have performed the gel filtration assays, and observed that in the homodimer size (150KDa) fraction, PDK1 was accumulated with 14-3-3 γ . Meanwhile, pPDK1(pS549) can also be detected at the same fraction (**revised Fig. 4j**). This result at least could partially indicate the proposal of pPDK1-(14-3-3)-pPDK1 to form complex (**revised Fig. 4n**).

11) Figure 5: Quantification, labelling. This is interesting information.

Response: As kindly suggested, we have quantified the key western blots (**revised Fig. 5e, 5h**) and IF images (**revised Fig. 5f**).

12) Figure 6: B: at the limit of resolution. Please adapt statistics. For F provide point by point statistical analysis (non-parametric) and 2-way ANOVA for the whole curve.

Response: As kindly suggested, we adjusted the Fig. 6b to improve the resolution. We also generated the point by point statistical analysis and 2-way ANOVA for the whole curve for **revised Fig. 6f**.

13) Figure 7: please integrate the established feedback loops emerging from S6K1.

Response: As kindly suggested, we have integrated the feedback loops in the **revised Fig. 7b** to make it more accurate.

**Minor stuff,
L300: mTROC2: typo**

Response: As kindly suggested, we have corrected it.

Use “cellular” where experiments were done in cells, use in vivo for results from tumor models

Response: As kindly suggested, we have corrected our findings to in cells or in mouse models.

Define ALL abbreviations

Response: As kindly suggested, we have defined all abbreviations in the revised manuscript.

Reviewer #3 (Remarks to the Author):

While it has long been appreciated the major role of AKT activation in cancer, upstream regulation of PDK1, which phosphorylates AKT at T308 is not fully understood. The authors identify a novel inhibitory phosphorylation site on PDK1 (S549) by S6K1 and propose a mechanism by which phosphorylation of this site decreases PDK1 membrane localization and allows 14-3-3 γ binding to promote its dimerization and prevent further activity towards AKT T308. They show mutations in and around this site promote tumorigenicity in cells and animals.

These studies are quite thorough and do a good job of demonstrating the PDK1 S549 is important for modulation of PDK1 activity. However, there are a few additional or updated figure panels will support your hypothesis more fully.

Response: We thank the reviewer in recognizing the novelty and the potential impacts of our study. We also appreciate the constructive comments from the reviewer to help us to further strengthen our manuscript.

Major comments:

1. As the p-PDK1 S549 antibody still binds with PDK1 S549A, please show validation of this antibody using a phosphatase with WT PDK1 to show decreased signal.

Response: As the reviewer mentioned, sometimes there is also some background for pS549-PDK1 could be detected with our generated pS549-PDK1 specific antibody. To further verify the specificity of this antibody, following the reviewer's kind suggestion, we have treated the cell lysates with λ protein phosphatase and found that λ -phosphatase treatment could dramatically attenuate the phosphorylation of PDK1-S549 compared with untreated cells (**revised Fig. S3d**). Thus, in combination with other findings (**revised Fig. 1b-h, S2b-c, S3b-f**), we conclude that this pS549-PDK1 antibody specifically recognizes the pS549-PDK1 event undergoing in cells.

2. Please add in an experiment where P551A mutant and 14-3-3 γ KD are done at the same time and cells are stimulated with insulin to show kinetics of AKT T308 phosphorylation under these conditions.

Response: Since P551A could largely decrease the interaction of PDK1 with 14-3-3 (**revised Fig. 4e, S6h**), we speculated that depletion of 14-3-3 γ could only mildly enhance PDK1-P551A-induced AKT phosphorylation. As the reviewer kindly suggested, we knockdown 14-3-3 γ in PDK1-P551A stably expressing cells, and stimulated the cell with insulin in a time course.

Compared with PDK1-WT and P551A, knockdown *14-3-3 γ* in P551A cell could mildly elevate AKT phosphorylation (**Fig. R1**), indicating that *14-3-3 γ* mainly through binding to the phosphorylated motif containing P551 to regulate the PDK1-AKT axis.

Fig. R1 Depletion of *14-3-3 γ* only mildly affects P551A functions in activating AKT phosphorylation. HEK293T-sgPDK1 cells expressed indicated encoding constructs or infected with shRNA against *14-3-3 γ* were serum-starved for 12 hrs, and stimulated with insulin in a time course manner, and subjected for IB analysis.

3. Nearly every Figure Legend is missing some very relevant information (N, how data was normalized). Interestingly, the authors did n = 5 mouse tumors for the growth assays but did not stain all 5 tumors for IHC or run all 5 on WB. I would like to see the data for all 5.

Response: As kindly suggested by the reviewer, we have added the relevant information in the Figure legend. As kindly suggested, we have included all the 5 tumors derived from 5 mouse expressing EV, WT, S549A, S549D-PDK1 for WB analysis (**Fig. R2**). Furthermore, we have obtained the similar IHC staining for 5 tumors (**revised Fig. 2i, S4g**).

Fig. R2 Tumors derived from 5 mouse bearing DLD1-PDK1^{-/-} cells expressed EV, WT, S549A, S549D-PDK1 were harvested for IB analysis. The results showed that S549A enhanced, whereas S549D decreased PDK1 function in promoting AKT-pT308.

4. Figure 7 Legend – Please add more information about the meaning of red vs blue arrows, solid vs dotted lines, and a more comprehensive explanation of the pathways shown.

Response: As kindly instructed, we have included the meaning of red vs blue arrows (red indicates activated function, while blue indicates repressed function). Furthermore, the solid lines indicate direct phosphorylation, whereas dotted lines indicate indirect phosphorylation. In the revised figure legends, we have comprehensively explained the pathways.

Minor comments:

5. Please check grammar and tense of verbs used throughout the publication, especially usage of the word “could”.

Response: As kindly instructed, we have removed almost all the word “could”, and further carefully edited the language and grammatical errors to make it easy to be followed.

6. Figure S1B – WT-PDK1 appears to be in the nucleus in this panel but not in S1C/E. Is this a representative image or can you choose another cell to better show the localization?

Response: As kindly suggested, we have repeated these staining, and obtained consistent findings which showed that PDK1 was mainly localized in the cytoplasm under normal condition or treated with the PI3K specific inhibitor BKM120 (**revised Fig. S1e**). On the other hand, PDK1 stimulated with insulin or harbored non-phosphorylation mutant S549A or other patients derived mutations including R544K, R546P, S549N, P551A, P551L, P551Q displayed markedly elevation of membrane locations, which has been further quantified for the plasma membrane to cytosol ratio (**revised Fig. 3h, 5f, S1d-e, S1g, S6f**).

7. Fig. 1- E) Show HA blot. F) S6K1 is missing its “1”. G) Please show an IgG control. “IB:PDK1” should read “IP:PDK1”.

Response: As kindly suggested, we have included HA blot in **revised Fig. 1e**, and corrected the mislabel of S6K1 in **revised Fig. 1f**. Furthermore, we have repeated the experiment in **revised Fig. 1g** included an IgG IP as control, meanwhile we have changed the typo IB: PDK1 to IP: PDK1 in this figure.

8. Line 128 – Insulin did enhance PDK1 phosphorylation in S6k1/2-/- MEFs. Please reflect this in the text.

Response: As kindly suggested, we mentioned this as “insulin could not strongly accelerate PDK1 phosphorylation in S6k1/2-/- MEFs ...” in the revised manuscript.

9. Line 131-132 – “depletion of S6K1 in cancer cells could also reduce PDK1 phosphorylation” -please cite this data

Response: As kindly suggested, we cite this data in the revised manuscript as **revised Fig. 1g**.

10. Line 135 – The S549A mutant form had reduced phosphorylation (Fig. 1I), not completely abrogated as the text states.

Response: As kindly suggested, we described this result more accurately in the revised manuscript as “S549A substitution has reduced S6K1-mediated PDK1 phosphorylation”.

11. Fig. S2A,B – Please show a lower exposure IP: Flag; IB:Flag blot (A) and WCL IB:Flag blot (B).

Response: As kindly suggested, we have showed the lower exposure of IP: Flag and IB: Flag blot (A) and WCL IB: Flag blot (B) (**revised Fig. S2a-b**).

12. Fig. 2A – The text says that AKT p-T308 is significantly elevated. Please show quantitation of this.

Response: As kindly suggested, we have quantified the pT308-AKT (**revised Fig. 2a**), indicating that S549A displayed significantly elevated AKT p-T308.

13. Fig. 2B – Can you run the WT and SA mutant on the same blot so we can accurately compare their signaling?

Response: We thank the reviewer for this excellent point. Actually, we have transferred the two gels into the same membrane, which has been further subjected for IB analysis to keep the same

conditions and exposure (**revised Fig. 2b**). By the way, we also run these proteins in the same gel and obtained similar results in different cell lines (**Fig. S4b, R3**).

Fig. R3 S549A mutant could enhance PDK1-mediated AKT-pT308. DLD1-*PDK1*^{-/-} cells were stably expressed with WT and S549A-PDK1, the resulting cells were serum starved for 12hrs and subjected for insulin stimulation for a time course. The cell lysates were subjected for IB analysis with indicated antibodies.

14. Fig. 2G/I – Are any other points significantly different?

Response: As kindly suggested, we have performed statistical analysis point by point with student *t* test between EV/WT and EV/S549D, S549A/S549D and observed that WT but not S549D could induce tumor growth compared with control cells. Meanwhile, 549A could significantly enhance tumor growth compared with WT, S549D-PDK1 (**revised Fig. 2e/i**).

15. Line 168 – Add Fig. 2E to the parentheses here

Response: As kindly suggested, we added **revised Fig. 2e** to the parentheses.

16. Fig. 2 - Can you explain the discrepancy on p-AKT S473 seen in Fig. 2B versus Fig. 2H? Effect of in vivo tumor growth versus cell culture?

Response: Since the major function of PDK1 is to promote AKT-pT308, our studies mainly focused on AKT-pT308. Sometimes, there are some cross talks between AKT-pT308 and AKT-

pS473 under different conditions. In our result, we observed that AKT-pS473 was sometimes affected by *PDK1* mutations either at S549A or S549D (**revised Fig. 2b**) or mutations derived from patient (**revised Fig. 5h**) under cell culture conditions, however, which could not be observed in mouse studies bearing mutation expressing tumors (**revised Fig. 2h, 6h**). These findings indicate that possibly more signals *in vivo* such as tumor microenvironment likely affect AKT-pS473 than in cell culture experiments.

17. Can you discuss why WT PDK1 does not seem to show any significant enhancement in tumorigenicity or cell growth phenotypes over empty vector? (Ex. Fig. S4E,F and Fig. 6)

Response: We thank the reviewer for pointing out this outstanding question. Indeed, as we observed, just ectopic expression of PDK1-WT could not significantly enhance AKT-pT308 and cell growth phenotypes either in cell culture or in mouse studies over empty vector like the reviewer mentioned (**Fig. S4e,f and Fig. 6**). The reason behind is possible due to the fact that PDK1 expression-induced mild activation of AKT should be not sufficient to promote cell growth phenotypes, whereas, mutations of *PDK1*, either non-phosphorylation mimic mutant S549A or patients-associated mutations were stronger enough to activate AKT kinase and augment cell growth phenotypes.

18. Fig. S5 – Please use “N.S.” instead of #” for P>0.05

Response: As kindly instructed, we have used “N.S.” to instead of “#” in the **revised Fig. S5**.

19. Fig. S6H – Can you add another couple lanes to this experiment where you add one WT and one SA/SD? Can you also show 14-3-3 γ in this figure?

Response: As kindly suggested, we have re-done the experiments and found that S549D could promote PDK1 dimerization, while S549A abrogate PDK1 dimerization (**revised Fig. S6h**). Moreover, we also included 14-3-3 γ blot in this figure and observed that S549D but not S549A, could enhance 14-3-3 γ binding with PDK1 (**revised Fig. S6h**).

20. Fig 3G – Please add S6K1 total

Response: As kindly suggested, we have included the blot for S6K1 expression (**revised Fig. 3F**).

21. Fig. S7C/; Fig. 4G – Please add p-PDK1 blot to show if the P551A mutant can still be phosphorylated.

Response: As kindly suggested, we have included pS549-PDK1 in the figures mentioned by the reviewer (**revised Fig. S7c and 4g**), where we observed that P549A partially decreased PDK1 phosphorylation at S549, possibly due to the generation of specific phosphorylation antibody with the peptides containing P551. Thus, this site mutation could markedly reduce the specific phosphorylation antibody to recognize pS549 (**revised Fig. S8a, lane 2**), while the pan-AKT substrate antibody could detect the potential phosphorylation of S549 without the effects from P551 mutation (**revised Fig. S8a, lane 1**).

22. Fig. S7F – Please use lower exposure blot for GST.

Response: As kindly suggested, we have provided a lower exposure blot for GST-14-3-3 in the **revised Fig. S7f**.

23. Fig. 4E – Please show lower exposure of both Flag blots.

Response: As kindly suggested, we have provided a lower exposure blot for Flag-PDK1 for both IP and WCL in the **revised Fig. 4e**.

24. Fig. 4I – What cell type is this? This one AKT activation blot looks different than the others - why doesn't WT activate AKT T308 until 30 minutes?

Response: We used HEK293T-sg*PDK1* cell infected with different mutant form of PDK1 for this experiment. Due to starvation and cultured condition, WT-PDK1 expressing cells display a weaker activation of AKT in 30 min after insulin stimulation. Whereas, P551A and S549A mutations strongly promoted pT308-AKT in the earlier period of 10 mins (**revised Fig. 4i**).

25. Fig. 4J – Please add 14-3-3 γ to the PD blot.

Response: As kindly suggested, we have obtained the 14-3-3 γ blot to show the pull down result (**revised Fig. S4k**).

26. Line 254-255 – Fig. S7I doesn't support your statement that 14-3-3 γ significantly enhances the interaction. Please quantify this over several experiments.

Response: As kindly instructed, we have showed a lower exposure of Fig. S7I and further quantified the PDK1 blot with ImageJ (**revised Fig. S7i**). From this data, we can conclude that depletion of 14-3-3 could significantly enhance PDK1 interaction with PIP₃.

27. Fig. 5 – Please continue the Type I and Type II labels throughout the figure. Please add Type I mutants to panel D. Add another experiment where you pulldown Flag and probe for 14-3-3 γ (opposite pulldown to 5C).

Response: As kindly suggested, we have provided the experiments including the Type I and Type II labels throughout the figure (**revised Fig. 5b-f**). We have also added Type I mutants to panel D (**revised Fig. S8a**), and performed the experiments to pulldown Flag and probed 14-3-3 γ (opposite pulldown to 5C) (**revised Fig. S8c**).

28. Fig. 5F – please quantify the “enhanced interaction” between AKT and PDK1 or mention how reproducible this is.

Response: As kindly suggested, we quantified the levels of HA-AKT1 and PDK1 in **revised Fig. 5g**, and the result showed that indeed these patients associated mutant *PDK1* could enhance PDK1/AKT interaction (**revised Fig. 5g**). We have repeated this experiment twice as we have mentioned in the revised method section.

29. Fig. 5G – Show lower exposure for pT308-AKT and add TI and TII labels. Both S549N and P551Q are not really convincing in their “enhanced AKT activation”. You may want to show other substrates of AKT to make this more convincing such as PRAS40.

Response: As kindly suggested, to the original Fig. 5g (**revised Fig. 5h**), we have provided a lower exposure for pT308-AKT and added the Type I and Type II labels (**revised Fig. 5h**). Furthermore, we have also quantified the pT308-AKT and included other AKT substrates such as pTSC2 and pPRAS40 (**revised Fig. 5h**).

30. Lines 267-269 – Please give more information about how frequent these mutations are in cancer, what types of cancer they occur in, do tumors ever have type I and type II together, and any other relevant information about this analysis.

Response: We thanks the reviewer for raising this great question. We agree that the description of how frequent the human *PDK1* mutations occur and which cancers these mutations are observed in is every important for future readers to comprehend our major conclusions. To this end, we summarized these mutations and observed that mutations mainly occurred in colon adenocarcinoma (**Fig. R4**). Thus, we chose a colon cancer cell DLD1 with an intact *PDK1*, among which *PDK1* has been genetically depleted as previously reported as a cell model (Kajsa Ericson, et al, *PNAS*, 2010). In this cell line, we re-introduced different *PDK1* patient-derived mutations including R544K, R546P, S549N and P551L, to investigate their oncogenic functions compared with WT *PDK1* (**revised Fig. 6**). To further validate our findings, we also chose another colon cancer cell line SW480 to study the potential oncogenic function of these *PDK1* mutations and obtained a similar result (**revised Fig. S8d-e**). These newly obtained data together suggest that *PDK1* mutations at least in colon cancer could promote cell malignancies partially by activating AKT kinase.

TCGA-EM-A2CJ...	Papillary Thyroid Cancer	P551Q	○	Missense
TCGA-FS-A1Z4-06	Cutaneous Melanoma	P551Q	○	Missense
PECAPJ41CLON...	Esophageal Poorly Differentiated ...	P551L	○	Missense
PECAPJ41CLON...	Esophageal Poorly Differentiated ...	P551L	○	Missense
TCGA-G2-A2EO...	Bladder Urothelial Carcinoma	P551A	○	Missense
coadread_dfci_2...	Colorectal Adenocarcinoma	S549N	○	Missense
P-0006793-T01-I...	Lung Adenocarcinoma	R546P	○	Missense
P-0004795-T01-I...	Colon Adenocarcinoma	R544K	○	Missense

Fig. R4. The mutant information for *PDK1* in different cancer types and its frequency by analyzing the TCGA database (<http://www.cbioportal.org/public-portal/>). As indicated, two Type I mutations including R544K, S549N were observed in colorectal adenocarcinoma, thus, we chose colon cancer cell lines including DLD1 and SW480 with intact *PDK1* to further investigate mutant *PDK1* oncogenic functions.

31. Line 326-328 – Please elaborate on why you think 14-3-3t and pS241-PDK1 might have an effect on the 14-3-3/pS549 interaction you demonstrate in this paper or delete this postulation.

Response: Since we have no more evidence to confirm the correlation and crosstalk of 14-3-3t/pS241-PDK1 affecting the 14-3-3γ/pS549, as kindly suggested, we have deleted this postulation in the revised manuscript, and would like to investigate this in the near future in a separate manuscript as a following up study.

Reviewers' comments:

Reviewer #1 (Remarks to the Author):

The authors have adequately addressed some of the criticisms, but unfortunately there are still several aspects of the manuscript that do not meet the necessary scientific standards for publication. In particular, all immunofluorescence and immunoblotting experiments are only quantified from one experiment and lack statistical analysis.

1. Although the authors have provided some further data regarding the PDK1 patient mutations in the rebuttal, there is still insufficient detail in the manuscript. The citation provided by the authors for TCGA (PMID: 25691825) is for a review rather than the original cohort study (PMID: 23000897). The authors also incorrectly state that all mutations are from the TCGA cohort. Only P551A and P551Q mutations are from TCGA cohorts, whereas S549 is from the DFCI cohort (PMID: 27149842) and R544K and R546P are from MSK-IMPACT cohorts (PMID: 28481359). The P551L mutation listed here is not from a patient tumor but rather an oral epithelial cancer cell line, PE/CA-PJ41 (clone D2), identified in the Cancer Cell Line Encyclopedia (PMID: 22460905, 31068700), and this needs to be indicated as such. For each mutation, the authors need to specify in the manuscript: a) the cancer type b) the cohort name with correct citation to original study c) size of cohort, and number/percentage of cases from this cohort that contain the specific mutation. This should be either indicated in the text, or included as a separate figure panel. See PMID: 31699932 as an example of how this should be presented.

2. The authors have added some densitometry analysis for the AKT signaling blots presented in the manuscript, but this analysis is only from one replicate and needs to be performed from 3 independent experiments with statistical analysis.

3. The authors claim they have calculated the plasma membrane to cytosolic ratio of PDK1 in Fig. 3h, 5f, Supplementary Fig. 1d, 1e, 1g, 6f as requested. However, the figures indicate "percentage of cells with PDK1 in the membrane", without stating how many cells or replicates, or including error bars and statistical analysis. There is no description in the methods section for how these immunofluorescence assays were performed, and how the analysis was conducted. This sounds like the authors have counted cells that appear to have membrane (is this plasma membrane?) PDK1 staining. This kind of qualitative analysis is subject to user bias, the authors need to use algorithm-based quantitative analysis in order to calculate the fluorescence intensity of plasma membrane versus cytosolic PDK1 from 3 independent experiments, and perform statistical analysis. See PMID: 26267533 for an example of how this analysis should be performed. The authors need to indicate either in the figure legend or method section exactly what analysis was performed, and how many cells were analyzed from each replicate.

4: The authors have amended the methods to state that "ubiquitination, soft agar, colony formation, glucose uptake and lactate production assays were performed twice independently with similar results". Colony formation assays and soft agar assays (Fig. 2c, 2d, 6b-e, Supplementary Fig. 4c, 5b, 5c, 7e, 9b) must be quantified from 3 independent experiments.

5. Fig. 6i and 6j are still expressed as "relative Ki67/pS6 positive numbers" rather than as percentage of Ki67/pS6 positive cells, please amend this.

Reviewer #2 (Remarks to the Author):

One of my major concerns - the question of reproducibility - was unfortunately not addressed by the authors. Although the authors provided a quantification of some panels in immunoblots, no repetitive data, means \pm SD, statistical analysis, phosphoprotein/loading control ratios and graphs analyzing these data were presented. The data therefore remain very preliminary. For a correct non-parametric analysis at least 6 reference/control experiments and 3 "challenged"

experimental values (biological replicates) are needed.

In this respect, a connection between the proposed mechanism (S6K-mediated PDK1 inhibition by S549 phosphorylation and the biological data presented in Fig. 6 cannot be made. This is disappointing, as the working hypothesis of this work is highly interesting.

Replay to authors rebuttal:

1) the provided analysis is insufficient (immunoblots, imaging data, etc.). All phospho-protein bands have to be normalized to loading controls. For independent biological sample numbers see above.

2) Statistics and analysis on phospho-proteins insufficient. Some improvement on the analysis of biological data. But the connection of mechanism and biological output remains weak. The time curves of the phospho-proteins should be analyzed and correlated with each other: how does Akt, S6K, etc. activation relate in a temporal fashion to S549 phosphorylation?

3) The main layout of the figures has not been improved.

4) I cannot deduce any quantitative information from Fig. S1a. In S33 it is not clear how the provided curves were generated, as there appear only to be 7 (n=1 ?) experimental points for stimulated cells. Reproducible data of such curves should be matched with pAKT, pS6K, pS6, etc. f(t). Normalized to controls. For S2c: no quantification provided (n=1).

5) Fig. R1 would rather contradict the proposed mechanism of the authors: a consistent activation of PI3K results in pS549, but AKT activation is not impaired. And BKM120 immediately eliminates pS549. This does not invalidate all of the claims of the authors, but clearly shows that more experiments are needed to understand the importance and timing of the S549 phosphorylation.

6) Some simple experiments with PDK1-CAAX or myr-PDK1, myr-Akt, etc. could have been tried. The use of PIKfyve inhibitors, endo-membrane disruptors, etc. might have provided some data on the site of phosphorylation.

7) Although some inhibitors were added, the presented data cannot be analyzed (n=1).

8) Same as above. Quantification of one blot is insufficient. Presentation of histology somewhat improved.

9) In spite of the values: n=1

10) More data (chromatograms), correlations, another cytosolic marker segregating from 14.3.3 would have to be shown to warrant a PDK1-14.3.3 complex.

11) As above. n=1

12) Improved. Query answered.

13) Figure 7 still needs work.

Reviewer #3 (Remarks to the Author):

The revisions are satisfactory. Please just address how the cyto/PM quantification was done in the Methods

Point by Point Rebuttal

(Manuscript number: NCOMMS-21-18003)

We sincerely appreciate the thorough analyses and constructive comments provided by the three reviewers, which have been very helpful in guiding us to further improve our manuscript. As described in detail below, we have fully addressed the concerns raised by the reviewers.

Reviewers' comments:

Reviewer #1 (Remarks to the Author):

The authors have adequately addressed some of the criticisms, but unfortunately there are still several aspects of the manuscript that do not meet the necessary scientific standards for publication. In particular, all immunofluorescence and immunoblotting experiments are only quantified from one experiment and lack statistical analysis.

Response: We thank the reviewer in recognizing the effort of our previously revised work. We also appreciate the constructive comments from the reviewer to help us to further strengthen our manuscript. In this round revision, we have mentioned that all the immunofluorescence and immunoblotting experiments were determined in triplicated. These data were quantified from three independent experiments with indicated statistical analysis as mentioned in figure legends.

1. Although the authors have provided some further data regarding the PDK1 patient mutations in the rebuttal, there is still insufficient detail in the manuscript. The citation provided by the authors for TCGA (PMID: 25691825) is for a review rather than the original cohort study (PMID: 23000897). The authors also incorrectly state that all mutations are from the TCGA cohort. Only P551A and P551Q mutations are from TCGA cohorts, whereas S549 is from the DFCI cohort (PMID: 27149842) and R544K and R546P are from MSK-IMPACT cohorts (PMID: 28481359). The P551L mutation listed here is not from a patient tumor but rather an oral epithelial cancer cell line, PE/CA-PJ41 (clone D2), identified in the Cancer Cell Line Encyclopedia (PMID: 22460905, 31068700), and this needs to be indicated as such. For each mutation, the authors need to specify in the manuscript: a) the cancer type b) the cohort name with correct citation to original study c) size of cohort, and number/percentage of cases from this cohort that contain the specific mutation. This should be either indicated in the text, or included as a separate figure panel. See PMID: 31699932 as an example of how this should be presented.

Response: We thanks the reviewer for raising this great question. Indeed, the detail information about PDK1 patient mutations will be include as the reviewer suggested. To this end, in this round revision, we have included the information as below:

- 1) We cited the original cohort study (PMID: 23000897) for the TCGA database in our revised manuscript.
- 2) We have corrected our state that P551A and P551Q mutations are from TCGA cohorts, S549 is from the DFCI cohort (PMID: 27149842) and R544K and R546P are from MSK-IMPACT cohorts (PMID: 28481359).
- 3) We thank the reviewer pointing out that the P551L mutation was derived from an oral epithelial cancer cell line, identified in the Cancer Cell Line Encyclopedia (PMID: 22460905), which we have cited in our revised manuscript.
- 4) As the reviewer kindly instructed, we have specified in the manuscript the following information in the text, such as a) the cancer type b) the cohort name with correct citation to original study c) size of cohort, and number/percentage of cases from this cohort that contain the specific mutation.

2. The authors have added some densitometry analysis for the AKT signaling blots presented in the manuscript, but this analysis is only from one replicate and needs to be performed from 3 independent experiments with statistical analysis.

Response: We thanks the reviewer for raising this critical comment. Since we have obtained these data from at least 3 independent experiments, with the reviewer kind suggestion, we have analyzed the major finding from 3 independent experiments with statistical analysis (**Revised Fig 1c-h, 2a-b, 2h, 3a-c, 3f, 4i, 4m, 5e, 5i, 5k, 6a, 6f, S2b, S2d, S2f, S3f, S8d**) for immunoblotting, (**Revised Fig 3g, 5g, S1d, S1e, S1g, S6f**) for immunofluorescence.

3. The authors claim they have calculated the plasma membrane to cytosolic ratio of PDK1 in Fig. 3h, 5f, Supplementary Fig. 1d, 1e, 1g, 6f as requested. However, the figures indicate “percentage of cells with PDK1 in the membrane”, without stating how many cells or replicates, or including error bars and statistical analysis. There is no description in the methods section for how these immunofluorescence assays were performed, and how the analysis was conducted.

This sounds like the authors have counted cells that appear to have membrane (is this plasma membrane?) PDK1 staining. This kind of qualitative analysis is subject to user bias, the authors need to use algorithm-based quantitative analysis in order to calculate the fluorescence intensity of plasma membrane versus cytosolic PDK1 from 3 independent experiments, and perform statistical analysis.

See PMID: 26267533 for an example of how this analysis should be performed. The authors need to indicate either in the figure legend or method section exactly what analysis was performed, and how many cells were analyzed from each replicate.

Response: As the reviewer kindly instructed, based on the paper the reviewer suggested, we performed the staining with 3 independent experiments, and counted more than 60 cells/experiment from three independent experiments to calculate the ration of PDK1 location in

plasma membrane or cytosolic which we have mentioned in the figure legends and method section of this revised manuscript. Briefly, the average pixel fluorescence intensity within an area of defined size drawn over 3 distinct areas of the plasma membrane or the cytosol were measured, and displayed as ratios of plasma membrane to cytosolic pixel fluorescence intensity. For each detection, at least 60 cells were calculated from 3 independent experiments. We have also cited the paper as the reviewer mentioned (PMID 26267533) in our revised manuscript.

4: The authors have amended the methods to state that “ubiquitination, soft agar, colony formation, glucose uptake and lactate production assays were performed twice independently with similar results”. Colony formation assays and soft agar assays (Fig. 2c, 2d, 6b-e, Supplementary Fig. 4c, 5b, 5c, 7e, 9b) must be quantified from 3 independent experiments.

Response: We are sorry for this misstatement. Actually, we have performed all these experiments three independently in triplicated with similar results. Colony formation assays and soft agar assays (Revised Fig. 2d, 2e, 6b-c, Supplementary Fig. 4c, 5b, 5c, 8e, 9b) have been quantified from 3 independent experiments as we mentioned in the figure legends. Moreover, we also analyzed all the major finding from 3 independent experiments with statistical analysis (Revised Fig 1c-h, 2a-b, 2h, 3a-c, 3f, 4i, 4m, 5e, 5i, 5k, 6a, 6f, S2b, S2d, S2f, S3f, S8d) for immunoblotting, (Revised Fig 3g, 5g, S1d, S1e, S1g, S6f) for immunofluorescence.

5. Fig. 6i and 6j are still expressed as “relative Ki67/pS6 positive numbers” rather than as percentage of Ki67/pS6 positive cells, please amend this.

Response: As the reviewer kindly suggested, we have re-analyzed the staining results with the label of “percentage of Ki67/pS6 positive cells” in the revised manuscript (Revised Fig. 2l, S4g, S8k, S8m).

Reviewer #2 (Remarks to the Author):

One of my major concerns - the question of reproducibility - was unfortunately not addressed by the authors. Although the authors provided a quantification of some panels in immunoblots, no repetitive data, means \pm SD, statistical analysis, phosphoprotein/loading control ratios and graphs analyzing these data were presented. The data therefore remain very preliminary.

For a correct non-parametric analysis at least 6 reference/control experiments and 3 "challenged" experimental values (biological replicates) are needed.

In this respect, a connection between the proposed mechanism (S6K-mediated PDK1 inhibition by S549 phosphorylation and the biological data presented in Fig. 6 cannot be made. This is disappointing, as the working hypothesis of this work is highly interesting.

Response: We appreciate the constructive comments from the reviewer to help us to further strengthen our manuscript. In this round revision, we have performed and mentioned that all the immunofluorescence and immunoblotting experiments were determined in triplicated. These data were quantified from three independent experiments with indicated statistical analysis as mentioned in the figure legends.

Actually, we have performed all these experiments at least three independently in triplicated with similar results. Colony formation assays and soft agar assays (**Revised Fig. 2d, 2e, 6b-c, Supplementary Fig. 4c, 5b, 5c, 8e, 9b**) have been quantified from 3 independent experiments as we mentioned in the figure legends. Moreover, we also analyzed all the major finding from 3 independent experiments with statistical analysis (**Revised Fig 1c-h, 2a-b, 2h, 3a-c, 3f, 4i, 4m, 5e, 5i, 5k, 6a, 6f, S2b, S2d, S2f, S3f, S8d**) for immunoblotting, (**Revised Fig 3g, 5g, S1d, S1e, S1g, S6f**) for immunofluorescence.

Replay to authors rebuttal:

1) the provided analysis is insufficient (immunoblots, imaging data, etc.). All phospho-protein bands have to be normalized to loading controls. For independent biological sample numbers see above.

Response: As the reviewer kindly suggested, we have performed all these experiments at least three independently in triplicated with similar results. Colony formation assays and soft agar assays (**Revised Fig. 2d, 2e, 6b-c, Supplementary Fig. 4c, 5b, 5c, 8e, 9b**) have been quantified from 3 independent experiments as we mentioned in the figure legends. Moreover, all phospho-protein bands have been normalized to loading controls from 3 independent experiments with accordingly statistical analysis (**Revised Fig 1c-h, 2a-b, 2h, 3a-c, 3f, 4i, 4m, 5e, 5i, 5k, 6a, 6f, S2b, S2d, S2f, S3f, S8d**) for immunoblotting, (**Revised Fig 3g, 5g, S1d, S1e, S1g, S6f**) for immunofluorescence.

2) Statistics and analysis on phospho-proteins insufficient. Some improvement on the analysis of biological data. But the connection of mechanism and biological output remains weak. The time curves of the phospho-proteins should be analyzed and correlated with each other: how does Akt, S6K, etc. activation relate in a temporal fashion to S549

phosphorylation?

Response: We thank the reviewer for this important comment. As the reviewer kindly suggested, we have included more statistics and analysis on phospho-proteins (**Revised Fig 1c-h, 2a-b, 2h, 4i, 4m, 5k, 6a, 6f, S2b, S2d, S2f, S3f, S8d**). The connection of mechanism and biological output was further validated with cell biological and *in vivo* mouse studies (**Revised Fig 2, 6, S4, S5, S8, S9**). Moreover, the time curves of the phospho-proteins have been analyzed and correlated with each other (**Revised Fig. S3e**). And we observed that pS549-PDK1 was correlated with pS6K and pS6, a well-established S6K substrate, meanwhile, pS6 and pS549-PDK1 were observed in a little later time of pAKT upon insulin/EGF stimulation (**Revised Fig. S3e**).

3) The main layout of the figures has not been improved.

Response: As the reviewer kindly instructed, we have re-designed the figure labelling and included the explanation of abbreviations in figure legends. Briefly, we have removed the repetitive labels and put important labels on the left of images to make the reading from left to right in our revised manuscript.

4) I cannot deduce any quantitative information from Fig. S1a. In S3e it is not clear how the provided curves were generated, as there appear only to be 7 (n=1 ?) experimental points for stimulated cells. Reproducible data of such curves should be matched with pAKT, pS6K, pS6, etc. f(t). Normalized to controls. For S2c: no quantification provided (n=1).

Response: We thank the reviewer raising this important comment. For Fig. S1a, the mass spectrometry analyses have been performed three times, and summarized in (**Revised Fig. S1b**), and quantified in (**Revised Fig. S1c**). From the quantified data, we can conclude that S549 phosphorylation occurs around 18%, whereas the auto-phosphorylation of S241 occurs around 80%, indicating the dynamic regulation of S549 and relatively stable status of S241 phosphorylation.

In S3e, we have performed these experiments three times with 7 experimental points for insulin or EGF stimulation respectively. The phosphorylation of pAKT, pS6K and pS6 were also included (**Revised Fig. S3e, top panels**), and normalized with according proteins and quantified with three reproducible results (**Revised Fig. S3e, bottom panels**).

For S2c, these results were representative of three independent experiments. The phosphorylation of S549 was normalized with immunoprecipitated PDK1, quantified (**Revised Fig. S2g, right panel**) and analyzed statistically with student *t* test.

5) Fig. R1 would rather contradict the proposed mechanism of the authors: a consistent activation of PI3K results in pS549, but AKT activation is not impaired. And BKM120 immediately eliminates pS549. This does not invalidate all of the claims of the authors, but

clearly shows that more experiments are needed to understand the importance and timing of the S549 phosphorylation.

Response: It is well established that a consistent activation of PI3K results in PIP3 accumulation, in turn recruiting PDK1 to the membrane for AKT phosphorylation and activation (Alessi *et al. Curr Biol*, 1997; L. Stephens *et al. Science*, 1998; D. Stokoe *et al. Science*, 1997). Based on these reports, although pS549-PDK1 and other S6K-mediated negative feedback regulation kept a relatively high level due to AKT-mediated S6K1 phosphorylation, PIP3-mediated AKT activation is only partially repressed and still remains in a relative high level. This potential mechanism could at least partially explain the reason of high pS549-PDK1 coupled with high AKT kinase activity upon PI3K gain-of-function mutations or PTEN deficiency.

To further study whether BKM120 immediately eliminates pS549-PDK1, we treated PI3K constitutively active cell T47D with a time course of PI3K inhibitor BKM120, and observed that pS549-PDK1 was decreased with the time until to 40 min, however, the phosphorylation of AKT and S6K were decline as early as 10 min after treated with BKM120 (Fig. R1). This indicates that upon treatment with BKM120, PIP3 deprivation would rapidly attenuate AKT and its downstream S6K phosphorylation, which would retard pS549-PDK1.

Fig R1, IB analysis of WCL and IP products derived from T47D cells treated with PI3K inhibitor BKM120 (10 μ M) for the indicated time period.

6) Some sir
been tried. The use of PIKfyve inhibitors, endo-membrane disruptors, etc. might have provided some data on the site of phosphorylation. 2

Response: To further investigate the roles of S6K-mediated phosphorylation in affecting PDK1 membrane location and in turn AKT phosphorylation, with the reviewer kind suggestion, we employed membrane enforced location PDK1 (myr-PDK1) for further studies. We observed that myr-PDK1 containing S549A and S549D, displayed a similar activity toward phosphorylation of AKT and its downstream targets (**Revised Fig S6g**), indicating that S6K-mediated S549 phosphorylation is important for PDK1 membrane location and affects AKT phosphorylation. To validate the cellular compartment of S6K-mediated PDK1 phosphorylation, PIKfyve inhibitor was utilized, and we observed that both pS549-PDK1 and pAKT were marked declined (**Fig R2**). Moreover, cell fractions were separated, and we observed that pS549-PDK1 was occurred both in cytosol and plasma membrane, interestingly, constitutively active S6K1-R3A expression could promote membrane PDK1 phosphorylation at S549 (**Fig R3**). These data together suggest that one hand S6K1 phosphorylates PDK1 in plasma membrane to dissociate it from membrane location and AKT activation, on the other hand, S6K1 phosphorylates PDK1 in cytosol to block it membrane recruitment, both resulting in the repression of AKT kinase activity.

Fig. R2 HEK293 cells were transfected with indicated constructs. Resulting cells were serum starved for 12 hrs and treated with or without PIKfyve inhibitors before insulin stimulated for 15 min, and then subjected for IB and IP analyses.

Fig. R3 HEK293 cells were transfected with indicated constructs. The cell cytoplasm (Cyto.) and plasma membrane (Memb.) fractions were separated and subjected for IB and IP analyses.

7) Although some inhibitors were added, the presented data cannot be analyzed (n=1).

Response: As we have mentioned in figure legends, these experiments in Figure 1 have been performed 3 independent times, and the results were representative of three independent experiments, the relative quantification of pS549, RxxpS and pT308 were statistically analyzed (**Revised Fig. 1c-h, S2e, S2f, S2h**).

8) Same as above. Quantification of one blot is insufficient. Presentation of histology somewhat improved.

Response: With the reviewer kind suggestion, we have mentioned in figure legends that these experiments in Figure 2 have been performed 3 independent times, and the results were representative of three independent experiments, the relative quantification of pT308 were statistically analyzed (**Revised Fig. 2a, 2b, 2i**).

9) In spite of the values: n=1

Response: As we have mentioned in figure legends, these experiments in Figure 3 have been performed 3 independent times, and the results were representative of three independent experiments, the relative quantification for WB and IF were statistically analyzed (**Revised Fig. 3a-c, 3f-h**).

10) More data (chromatograms), correlations, another cytosolic marker segregating from 14.3.3 would have to be shown to warrant a PDK1-14.3.3 complex.

Response: As the reviewer kindly suggested, we have obtained another chromatogram data to show that ectopic expression of S6K-3R could enhance PDK1 complex formation compared with control group (**Fig. R4**). More interesting, S6K-3R expression also could potentially enhance 14-3-3 shift into with PDK1, coupled with its another segregating cytosolic marker YAP shift. However, the cytosolic marker Tubulin was not affected in our experiment settings.

Fig. R4 HEK293 cells were transfected with indicated constructs. and subjected for gel-filtration assays.

11) As above. n=1

Response: As we have mentioned in figure legends, these experiments in Figure 5 have been

performed 3 independent times, and the results were representative of three independent experiments, the relative quantification of pT308, HA-AKT1 and flag-PDK1 were statistically analyzed (**Revised Fig. 5e-1**).

12) Improved. Query answered.

Response: We thank the reviewer satisfied with our answers.

13) Figure 7 still needs work.

Response: We further optimized the model to emphasize the feedback regulation of mTORC1-S6K1 axis on PI3K-AKT pathway.

Reviewer #3 (Remarks to the Author):

The revisions are satisfactory. Please just address how the cyto/PM quantification was done in the Methods

Response: We thank the reviewer in recognizing the effort of our previously revised work. We also appreciate the constructive comments from the reviewer to help us to further strengthen our manuscript. In this round revision, we have included the method on the quantification of cyto/PM.

REVIEWERS' COMMENTS

Reviewer #1 (Remarks to the Author):

The authors have addressed the major concerns. They have provided sufficient detail regarding the PDK1 mutations in patient cohorts, and have performed quantification from at least three independent experiments as requested. However, the authors do not appear to have performed the correct statistical analysis of the data. In the methods the authors state that they have determined statistical significance by t test or two-way ANOVA, but in the figure legends only t tests are described. T tests are not appropriate for experiments in this manuscript as all data consists of more than two groups. Two-way ANOVA would only be appropriate for experiments with two independent variables, whereas one-way ANOVA should be used for experiments with one independent variable. This needs to be amended before publication.

Minor comments:

1. The authors should acknowledge in the discussion that the PDK1 mutations identified in this manuscript are very rare (<1% of cases), especially when compared to the other PI3K pathway mutations described in this section.
2. On page 13 last line, the authors state they analyzed the TCGA database, but this should be cBioPortal for Cancer Genomics.
3. The authors state, "Of note, type I, a much less extent of type II mutants" (page 14, line 12), please revise the grammar in this sentence.
4. In Figure 3g and 5g, the authors incorrectly refer to the representative immunofluorescence images as "graphic representation".
5. Supplementary Figure 4c is missing a legend.

Reviewer #2 (Remarks to the Author):

Thanks to the authors for their effort to document reproducibility and provide statistical analysis. Relevant data is now shown and analyzed at least as n=3 samples.

Figure quality and presentation was improved.

Figure 4n and Fig 7a, b, c still need work. The plasma membrane should not be presented as dotted line, but a broad line allowing the placement of a PIP3 label where the PH domain of the effector proteins / complexes can bind. For TORC2, the placement of the PH domain should be on a distinct SIN1 protein. Origin and targets of arrows should be better defined, currently start, shape and target of the arrows are unclear. Too much pathway abbreviation is confusing for the non expert reader, the arrow in 4n from Akt to S6K phosphorylation will be misunderstood. The quality of the arrows should be better defined: for example green solid lines with arrow for direct activations, green dotted lines with arrow for indirect (multistep) activation steps. Currently the color concept is confusing. Complement the PDK1 - phosphorylated or 14-3-3 bound with a status line (active, inactive).

Fig. 7: Remove "other kinases", straighten arrows, align TORC2, AKT, and PDK1 horizontally, number and name (single letter aa code) phosphorylated residues. Eliminate crossing arrows. Assign type of mutations with output signals (7c). Check consistency. IRS is not downstream of EGF (7b). When PDK1 is mutated, the S6K to IR->IRS negative feedback loop is still in place. Under "normal conditions" in the presence of Ins stimulation the feedback in 7b is also in place. Fig. 7 is just now not very helpful to illustrate how a changed PDK1 activity would shift the flow across the IR->PI3K->mTOR pathway. Please revise.

Point by Point Rebuttal

(Manuscript number: NCOMMS-21-18003Z)

We sincerely appreciate the thorough analyses and constructive comments provided by the reviewers, which have been very helpful in guiding us to further improve our manuscript. As described in detail below, we have fully addressed the concerns raised by the reviewers.

Reviewers' comments:

Reviewer #1 (Remarks to the Author):

The authors have addressed the major concerns. They have provided sufficient detail regarding the PDK1 mutations in patient cohorts, and have performed quantification from at least three independent experiments as requested. However, the authors do not appear to have performed the correct statistical analysis of the data. In the methods the authors state that they have determined statistical significance by t test or two-way ANOVA, but in the figure legends only t tests are described. T tests are not appropriate for experiments in this manuscript as all data consists of more than two groups. Two-way ANOVA would only be appropriate for experiments with two independent variables, whereas one-way ANOVA should be used for experiments with one independent variable. This needs to be amended before publication.

Response: We thank the reviewer in recognizing the effort of our previously revised work. We also appreciate the constructive comments from the reviewer to help us to further strengthen our manuscript. In this round revision, with the reviewer's kind instruction, we have performed the correct statistical analysis of the data. In the figure legends, we have demonstrated that we used two-way ANOVA to analyze two independent variables in Fig. 2f, 4i, 4m, 6d, and other experiments were analyzed with t tests. Furthermore, we provided the exact P values for each statistics analysis, and mentioned "Statistical significance was determined by using two-tailed, one-sample t-test for treatment comparisons and two-way ANOVA test for group comparisons.

Results are shown as mean \pm SEM. Source data are provided as a Source data file” in the figure legends.

Minor comments:

1. The authors should acknowledge in the discussion that the PDK1 mutations identified in this manuscript are very rare (<1% of cases), especially when compared to the other PI3K pathway mutations described in this section.

Response: As the reviewer kindly suggested, we have acknowledged in discussion that compared with other PI3K pathway mutations including *PIK3CA*, *AKT* and *PTEN* mutations, *PDK1* mutations are very rare (<1% of cases).

2. On page 13 last line, the authors state they analyzed the TCGA database, but this should be cBioPortal for Cancer Genomics.

Response: We have corrected this, and changed the statement of database to cBioPortal for Cancer Genomics.

3. The authors state, “Of note, type I, a much less extent of type II mutants” (page 14, line 12), please revise the grammar in this sentence.

Response: We are sorry for this grammar error, we have corrected this to “a lesser extent of..”.

4. In Figure 3g and 5g, the authors incorrectly refer to the representative immunofluorescence images as “graphic representation”.

Response: As kindly instructed, we have corrected this in the figure legends as “representative immunofluorescence images”.

5. Supplementary Figure 4c is missing a legend.

Response: We have included the figure legends for supplementary Figure S4c.

Reviewer #2 (Remarks to the Author):

Thanks to the authors for their effort to document reproducibility and provide statistical analysis. Relevant data is now shown and analyzed at least as n=3 samples.

Figure quality and presentation was improved.

Response: We appreciate the constructive comments from the reviewer to help us to further strengthen our manuscript.

Figure 4n and Fig 7a, b, c still need work. The plasma membrane should not be presented as dotted line, but a broad line allowing the placement of a PIP3 label where the PH domain of the effector proteins / complexes can bind. For TORC2, the placement of the PH domain should be on a distinct SIN1 protein. Origin and targets of arrows should be better defined, currently start, shape and target of the arrows are unclear. To much pathway abbreviation is confusing for the non expert reader, the arrow in 4n from Akt to S6K phosphorylation will be misunderstood. The quality of the arrows should be better defined: for example green solid lines with arrow for direct activations, green dotted lines with arrow for indirect (multistep) activation steps. Currently the color concept is confusing. Complement the PDK1 - phosphorylated or 14-3-3 bound with a status line (active, inactive).

Response: With the reviewer's kind suggestion, we have optimized the models in Figure 4n and Fig 7a-c, including the change of plasma membrane and displaying PH domain of SIN1 protein. We also defined and currently start the arrows in these models and avoided these abbreviations. As kindly instructed, we also included the detail components between AKT and S6K phosphorylation. The arrow lines were further defined as that red solid lines with arrow for direct activations, red dotted lines with arrow for indirect (multistep) activation steps, black solid lines with arrow for direct inactivation. We also complemented the link between the PDK1 - phosphorylation and 14-3-3 binding with a status line, indicating PDK1 phosphorylation promoted 14-3-3 binding and PIP3 disassociation (Revised Fig. 4n).

Fig. 7: Remove "other kinases", straighten arrows, align TORC2, AKT, and PDK1 horizontally, number and name (single letter aa code) phosphorylated residues. Eliminate crossing arrows. Assign type of mutations with output signals (7c).

Check consistency. IRS is not downstream of EGF (7b). When PDK1 is mutated, the S6K to IR->IRS negative feedback loop is still in place.

Under "normal conditions" in the presence of Ins stimulation the feedback in 7b is also in place. Fig. 7 is just now not very helpful to illustrate how a changed PDK1 activity would shift the flow across the IR->PI3K->mTOR pathway. Please revise.

Response: We thank the reviewer's insightful suggestion. As suggested, we have re-drawn the model, and removed "other kinases", straighten arrows, align TORC2, AKT, and PDK1 horizontally, labelled number and name of phosphorylated residues, eliminated the crossing arrows, assigned type of mutations with output signals (Revised Fig 7).

As the reviewer mentioned, IRS is not downstream of EGF, thus here we not emphasized EGF. Upon growth factors (insulin/EGF) stimulation, more PIP3 were generation, which promotes the location of AKT, PDK1, and mTORC2 with their PH domain, in turn promoting AKT phosphorylation. As the first finding of S6K-mediated negative regulation of AKT, S6K phosphorylates IRS partially reduced PIP3 generation, in turn decreased membrane location of these proteins upon insulin stimulation. Later study indicates that S6K1-mediated Sin1 phosphorylation blocks Sin1 containing mTORC2 membrane location, thus declines AKT phosphorylation at S473. As a central kinase, regulation of PDK1 in influencing AKT kinase activity is rarely investigated. Here we show that S6K phosphorylates PDK1 and in turn recruits 14-3-3 to block PDK1 membrane location, and AKT phosphorylation at T308 (7b). As a result, when PDK1 is mutated, although S6K-mediated IR->IRS negative feedback loop is still in place, increased membrane location of PDK1 markedly contributes to Akt activation and oncogenic roles (7c).

In 7a, we only show the early period of insulin/EGF stimulation, which could activate PI3K-AKT-mTOR pathway to play physiological function. Indeed, as the reviewer point out, under "normal conditions" in the presence of insulin stimulation, the feedback in 7b is in place to

calm down the constitutive activation of this pathway. We hope the reviewer could agree with us that all these negative feedback loops integrate together to tightly regulate PI3K-AKT-mTOR signaling to perform physiological roles. While, the mutations of *PDK1* and other proteins in these feedback loops would break this negative feedback regulation, and constantly activates AKT kinase and its oncogenic roles (Revised Fig 7c).